# AIMP2-DX2 provides therapeutic interface to control KRAS-driven tumorigenesis

Dae Gyu Kim[1,7], Yongseok Choi[2,7], Yuno Lee[3], Semi Lim[1], Jiwon Kong [1], JaeHa Song[1], Younah Roh[1], Dipesh S. Harmalkar[2,4], Kwanshik Lee[2], Ja-il Goo[2], Hye Young Cho[5], Ameeq Ul Mushtaq[5], Jihye Lee[1], Song Hwa Park[1], Doyeun Kim[1], Byung Soh Min [6], Kang Young Lee[6], Young Ho Jeon[5], Sunkyung Lee[3], Kyeong Lee [4✉] & Sunghoon Kim [1✉]

Recent development of the chemical inhibitors specific to oncogenic KRAS (Kirsten Rat Sarcoma 2 Viral Oncogene Homolog) mutants revives much interest to control KRAS-driven cancers. Here, we report that AIMP2-DX2, a variant of the tumor suppressor AIMP2 (aminoacyl-tRNA synthetase-interacting multi-functional protein 2), acts as a cancer-specific regulator of KRAS stability, augmenting KRAS-driven tumorigenesis. AIMP2-DX2 specifically binds to the hypervariable region and G-domain of KRAS in the cytosol prior to farnesylation. Then, AIMP2-DX2 competitively blocks the access of Smurf2 (SMAD Ubiquitination Regulatory Factor 2) to KRAS, thus preventing ubiquitin-mediated degradation. Moreover, AIMP2-DX2 levels are positively correlated with KRAS levels in colon and lung cancer cell lines and tissues. We also identified a small molecule that specifically bound to the KRAS-binding region of AIMP2-DX2 and inhibited the interaction between these two factors. Treatment with this compound reduces the cellular levels of KRAS, leading to the suppression of KRAS-dependent cancer cell growth in vitro and in vivo. These results suggest the interface of AIMP2-DX2 and KRAS as a route to control KRAS-driven cancers.

[1] Medicinal Bioconvergence Research Center, Institute for Artificial Intelligence and Biomedical Research, College of Pharmacy & College of Medicine, Gangnam Severance Hospital, Yonsei University, Incheon, Korea. [2] Department of Biotechnology, Korea University, Seoul, Korea. [3] Drug Information Research Center, Korea Research Institute of Chemical Technology, Daejeon, Korea. [4] College of Pharmacy, Dongguk University, Goyang, Korea. [5] College of Pharmacy, Korea University, Sejong, Korea. [6] Department of Surgery, College of Medicine, Yonsei University, Seoul, Korea. [7] These authors contributed equally: Dae Gyu Kim, Yongseok Choi. ✉email: kaylee@dongguk.edu; sunghoonkim@yonsei.ac.kr

The three RAS genes encode four protein isoforms, KRAS4A, KRAS4B, NRAS, and HRAS, showing a high resemblance in their genetic sequences, protein structures and biochemical activities. Among these isoforms, KRAS mediates diverse cell signaling processes, and its mutations are frequently found in diverse cancers[1–3]. KRAS shuttles between guanosine triphosphate (GTP)-bound active and guanosine diphosphate (GDP)-bound inactive forms. This process is facilitated by GTPase-activating protein (GAP) and guanine nucleotide exchange factor (GEF), respectively. KRAS proteins consist of an N-terminal G domain and a C-terminal hypervariable region (HVR). The G domain contains switch I (SI) and switch II (SII) regions that are responsible for conformational changes during GDP-GTP exchange as well as for binding with downstream molecules, such as RAF proteins, phosphatidylinositol 3-kinase (PI3K), and Ral guanine nucleotide dissociation stimulator (RALGDS). While the C-terminal HVR is important for membrane trafficking, the isoforms are distinguished in their sequences. KRAS is farnesylated at the CAAX motif of HVR by farnesyl transferase for membrane localization and recruits GEF and GAP for its on and off transition, activating downstream molecules for cell proliferation.

Hyperactivating mutations in KRAS are found in 86–96% of pancreatic cancers, 40–54% of colorectal cancers (CRCs), and 27–39% of lung adenocarcinoma[4,5] and its mutations show a high degree of association with cancer progression and poor prognosis, prompting efforts to understand its role in tumorigenesis and identify drugs targeting KRAS[6–8]. Despite decades of research, direct targeting of oncogenic KRAS activity was challenging mainly because of the lack of an appropriate pocket for drug binding in the G domain of KRAS and its high affinity to the substrate. However, a recent trial to find ways to control oncogenic KRAS, AMG510, an inhibitor of the KRAS G12C mutant, was developed[9]. It irreversibly inhibits the KRAS G12C mutant by freezing it in an inert GDP-binding state via covalent binding to the pocket in the G domain[9]. While AMG510 was proven effective in the KRAS G12C mutant, controlling other KRAS mutants still remains to be solved[10]. In addition, an enhanced level of KRAS is considered a significant factor for cancer cell proliferation and tumorigenesis. In patients with gastric, breast, and head and neck cancers with no KRAS oncogenic mutations, KRAS overexpression is associated with poor prognosis[11–17]. Thus, an alternative route needs to be developed to broadly control KRAS-driven tumorigenesis, regardless of mutation type.

Aminoacyl-tRNA synthetase (ARS)-interacting multifunctional protein 2 (AIMP2) is known as a scaffold factor for the assembly of the multi-tRNA synthetase complex (MSC)[18]. Interestingly, when AIMP2 is dissociated from the MSC, it can act as a multifaceted tumor suppressor, controlling the p53[19], tumor necrosis factor (TNF)-α[20], transforming growth factor (TGF)-β[21,22], and Wnt[23] pathways depending on the cell context. AIMP2-DX2 (DX2 hereafter) was discovered as a splicing variant of AIMP2, which lacks exon II[24]. DX2 compromises the tumor-suppressive activities of AIMP2 by competing for the binding sites of AIMP2-binding proteins[24,25]. DX2 levels are positively correlated with tumor aggressiveness and poor prognosis in lung, ovarian, and colon cancers, and DX2 overexpression can transform normal cells to cancer[24,25], implying that DX2 is a potential target for cancer control. This potential has been validated by the suppression of DX2 expression with siRNA, shRNA, and small chemicals working on DX2 mRNA in various cancer cells and model systems[24–26].

DX2 was shown to be further stabilized at the protein level by heat shock protein 70 (HSP70), and the chemical inhibition of its interaction with HSP70 reduced DX2-induced tumor growth[27]. Interestingly, co-expression of DX2 increased the incidence of tumors in mice expressing the KRAS mutant[28], suggesting a potential connection between the two oncogenic proteins in tumor formation and growth. Here, we investigated the underlying molecular mechanism responsible for the synergistic effect of DX2 and KRAS on tumorigenesis and explored a route to control KRAS-driven cancers.

## Results

**DX2 enhances the protein stability of KRAS, but not of NRAS and HRAS.** To understand the functional relationship between DX2 and KRAS, we altered DX2 levels by ectopic expression of Strep-DX2 and si-DX2 in H460 lung cancer cells, and checked the change in KRAS levels. KRAS expression varied according to the cellular levels of DX2 at the protein level but not at the mRNA level (Fig. 1a). We further validated this result using colorectal tissues isolated from transgenic mice in which DX2 expression was induced by doxycycline (Dox) (see Methods for details). Induction of DX2 in transgenic mice led to a significant increase in KRAS levels (Fig. 1b). Dox-dependent induction of DX2 also led to an increase in KRAS levels in DX2-inducible mouse embryonic fibroblasts (MEFs) and H460 lung cancer cells stably expressing Strep-tagged DX2 (Fig. 1c and Supplementary Fig. 1a).

RAS proteins are encoded by the K, N, and HRAS isoforms, each of which undergoes specific cancer-related mutations[5,29] (Supplementary Fig. 1b). We determined the specificity of DX2 on the levels of RAS isoforms by expressing AIMP2 and DX2 in CCD18CO cells. The cells expressing DX2, but not AIMP2, increased the levels of both the wild-type (WT) and mutant forms of the KRAS, but not the N and H isoforms (Fig. 1d and Supplementary Fig. 1c). To determine the effect of DX2 on the stability of RAS proteins, we treated the cells with cycloheximide to block de novo protein synthesis and monitored the abundance of RAS isoforms over time. The stability of both WT and mutant KRAS was enhanced by the introduction of DX2 (Fig. 1d and Supplementary Fig. 1d). We also tested the effect of DX2 on ubiquitination of KRAS and found that DX2 reduced ubiquitination of KRAS but not N and H isoforms, implying that DX2 reduced KRAS ubiquitination (Fig. 1d and Supplementary Fig. 1e). After confirming that cell viability could be controlled by the expression level of KRAS (Supplementary Fig. 1f), we checked whether DX2 could influence RAS-induced cell proliferation and transformation[4,30] and found that DX2 further enhanced the KRAS, not N and H, -mediated cell proliferation and transformation (Fig. 1d and Supplementary Fig. 1g, h). We examined the effect of DX2 on the in vivo oncogenic activity of KRAS4B and found that overexpression of KRAS4B increased tumor growth, which was compromised by DX2 suppression (Fig. 1e and Supplementary Fig. 2a–c). These results suggest that DX2 stabilizes KRAS by blocking its ubiquitin-mediated degradation, promoting KRAS-induced cellular proliferation and transformation.

To determine the clinical relevance of the relationship between DX2 and KRAS, we analyzed their protein levels in various lung, colorectal, and pancreatic cancer cell lines expressing KRAS WT or oncogenic mutation[4,5] (Supplementary Table 1) and observed that the two protein levels were positively correlated in cancer cells, while they were expressed at low levels in normal lung and colorectal cell lines (WI-26 and CCD18CO, respectively) (Fig. 1f). Next, we investigated the correlation of the two proteins in the tumor tissues of patients with lung and colon cancers by immunohistochemistry (IHC) staining after checking the validity of KRAS and DX2 antibodies[31] for immunostaining (Supplementary Fig. 2d), and found a positive correlation between DX2 and KRAS, as observed in cancer cell analysis (Fig. 1g). IHC staining of tissue microarrays of lung and colorectal cancer

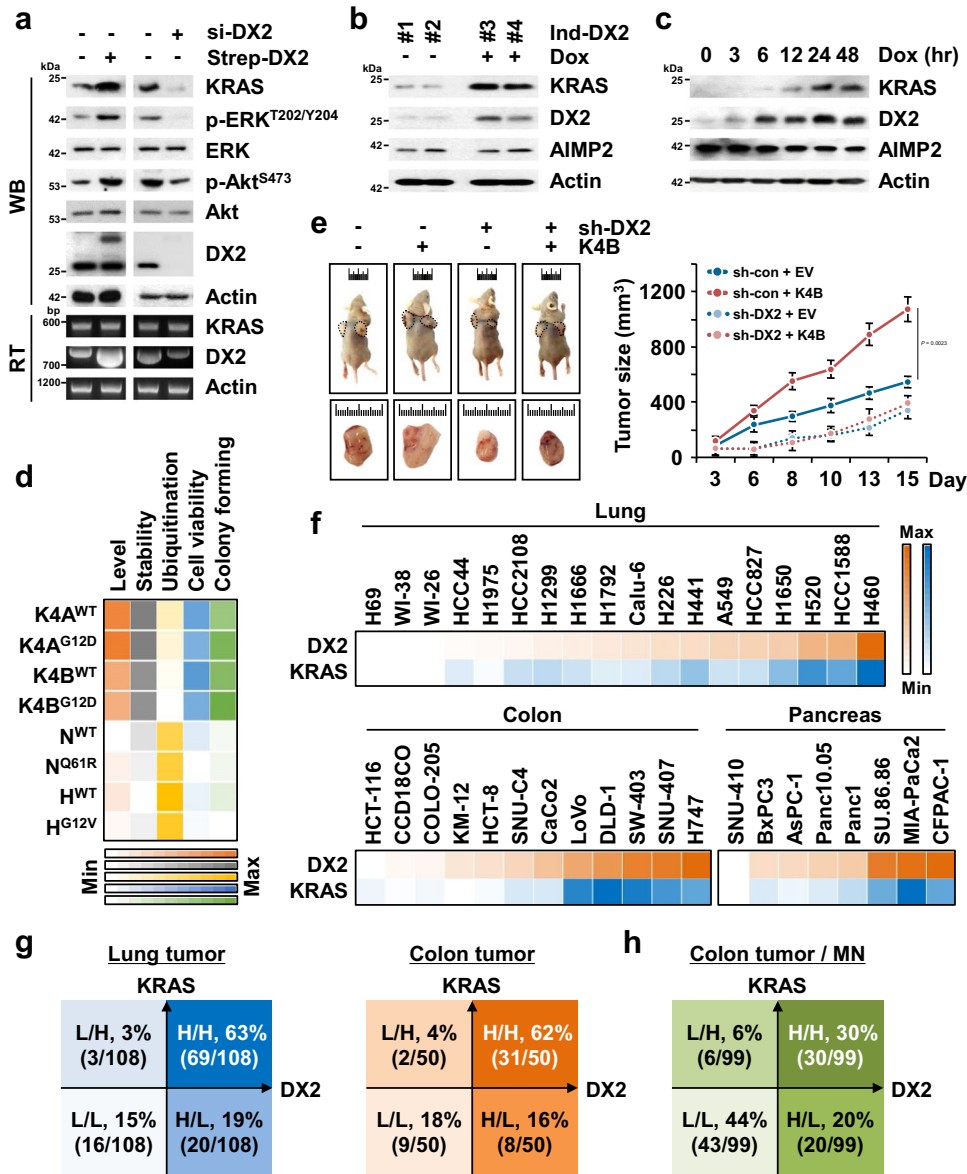

**Fig. 1 Enhancement of KRAS protein stability by DX2. a** DX2 levels in H460 lung cancer cells were controlled by the introduction of Strep-DX2 and si-DX2. Protein and mRNA levels were determined by Western blotting (WB) and RT-PCR (RT), respectively. Phosphorylation of ERK and Akt was monitored for KRAS signaling. Actin was used as a loading control. Results are representative of at least three independent experiments. **b** KRAS level in doxycycline (Dox)-inducible DX2 transgenic mice (Ind-DX2). Levels of KRAS and DX2 in colorectal tissues from two controls (#1, 2) and two Dox-fed mice (#3, 4) were analyzed. **c** MEF isolated from Ind-DX2 were treated with Dox in a time-dependent manner and protein levels were determined. Results are representative of at least three independent experiments. **d** Heat map from the results showing DX2-dependent regulation of RAS isoforms. DX2-mediated changes in the levels, stability, and ubiquitination of isoforms were quantified from Supplementary Fig. 1c–e. Cell viability and anchorage-independent colony formation assay were quantified from Supplementary Fig. 1g, h. Maximum (max) and minimum (min) values from the indicated experiments were designated as different colors with the highest and lowest intensity, respectively, and the rests were graded according to their relative values. **e** Result of xenograft ($n = 4$) using H460 cells stably expressing the indicated combination of KRAS4B and sh-DX2. Representative images of mice and tumors for each group (left) and tumor sizes (right) were shown. Sh means short hairpin RNA. All error bars represent the standard deviation (S.D.). $P$ value is from the two-sided $t$ test. **f** Heat map representing the cellular levels of DX2 and KRAS in 18 lung, 12 colorectal and 8 pancreatic cell lines. The cellular levels of DX2 and KRAS in the tested cell lines were represented as the heat map of orange and blue colors, respectively. The maximum and minimum values quantitated as described in Methods were shown with the highest and lowest color intensity, respectively, and the rests were graded according to their relative values. **g** Analysis of the levels of DX2 and KRAS in lung and colorectal tumor tissues. Staining intensities of the two proteins were classified as low (L) and high (H) (Supplementary Fig. 2e). Number of the analyzed tissue samples is shown in brackets. **h** DX2 and KRAS levels in the tumor and matched normal (MN) tissues from 99 patients with colorectal cancer. Depending on the levels in tumor compared to MN, samples were classified as H and L as compared to MN levels (Supplementary Fig. 2f). Source data are provided as a Source data file.

patient tissues showed that 63% of lung cancer ($n = 69/108$) and 62% of colon cancer ($n = 31/50$) expressed high levels of DX2 and KRAS proteins (Fig. 1g and Supplementary Fig. 2e). In contrast, only 15% and 18% of the lung and colon cancer tissues, respectively,

expressed low levels of both proteins. These data were further confirmed in matched tumor and normal tissues from patients with colorectal cancer ($n = 99$), where 30% ($n = 30/99$) of the patients expressed high levels of both proteins and 44% ($n = 43/99$)

expressed low levels (Fig. 1h and Supplementary Fig. 2f). These results suggest that DX2 and KRAS protein levels are tightly associated with some cancers.

**Specific interaction of DX2 with KRAS, but not with NRAS and HRAS.** To investigate the mechanisms by which DX2 enhances the stability of KRAS, DX2-interacting proteins in the presence of epidermal growth factor (EGF) signal, most significant for KRAS-driven cancer growth[32,33], were enriched by affinity purification and identified by liquid chromatography-mass spectrometry (Supplementary Fig. 3a). Among the 497 identified potential DX2 interactors, the top 200 proteins were classified by ontology (Supplementary Fig. 3b and Supplementary Table 2). The ontological analysis identified 12 proteins that are known to play roles in cell proliferation, and KRAS4A was revealed at the second highest frequency (Fig. 2a, see the box). Considering that the cells were treated with EGF, a known activator of KRAS[34–36], we focused on KRAS for further in-depth mechanistic analysis. We conducted in vitro pull-down and immunoprecipitation assays using DX2 with each of the K, N, and HRAS isoforms and validated the direct and specific interaction of DX2 with KRAS (Fig. 2b, c). We also confirmed the direct interaction of DX2 and KRAS4B, but not HRAS, by an in vitro pull-down assay using purified GST-DX2 and KRAS4B or HRAS proteins (Fig. 2d). After checking the capability of KRAS antibody for immunoprecipitation (Supplementary Fig. 3c), we examined the cellular interaction of endogenous KRAS and DX2 in colorectal adenocarcinoma DLD-1 cells and observed that the binding of the two proteins was enhanced by EGF treatment (Fig. 2e). In contrast to KRAS, peroxiredoxin 1 (PRDX1)[37], the most frequently detected protein among the DX2 interactors, interacted with DX2 independently of EGF (Supplementary Fig. 3d).

To analyze whether DX2 binds to cytosolic or membrane-anchored KRAS, we determined where the two factors would bind by monitoring endogenous DX2 and GFP-KRAS4B in 293 T cells with or without EGF signaling. Co-localization of the two proteins was significantly enhanced in the cytosol, but not in the plasma membrane, in the presence of EGF signal (Supplementary Fig. 3e), implying that DX2 would bind to KRAS prior to its membrane translocation. Surface plasmon resonance (SPR) using the purified KRAS4B and DX2 proteins estimated the $K_D$ value of the two proteins as 153 nM (Supplementary Fig. 3f). We mapped the protein domains responsible for the interaction of DX2 and KRAS4B by immunoprecipitation and in vitro pull-down assays, using the N-terminal flexible region (NFR) and GST domain of DX2 and the G domain and hypervariable region (HVR) of KRAS[27,29] (Supplementary Fig. 4a). The DX2 GST domain bound more strongly to the HVR than to the G domain of KRAS4B (Fig. 2f and Supplementary Fig. 4b, c). Given that the HVR sequence varies substantially between different RAS isoforms, these data are consistent with the isoform-specific binding of DX2 to KRAS[1,38] (Supplementary Fig. 4a and Fig. 2b, c). From the domain mapping results and comparison of the HVR sequence among RAS isoforms, KRAS HVR appears to be a crucial determinant for specific binding with DX2. We further confirmed the binding of KRAS HVR and DX2 by an in vitro pull-down assay using synthetic peptides and purified proteins. Biotinylated-KRAS4B, but not -HRAS, HVR peptide precipitated DX2 protein and native KRAS4B HVR peptide competitively removed the binding, indicating a specific interaction between DX2 and KRAS4B HVR (Fig. 2g). Although AIMP2 showed similar binding ability to KRAS4B as DX2 in in vitro binding assays because it also contains the GST domain (Supplementary Fig. 4d), it is unlikely to function as a KRAS stabilizer because it is mainly bound to the multi-tRNA

synthetase complex (MSC)[39]. In contrast, DX2 exists as an MSC-unbound free form[24], thereby being more accessible to KRAS for stabilization. To further validate this possibility, CCD18CO cells expressing both DX2 and AIMP2 were incubated in the absence and presence of EGF, and the protein extracts were subjected to gel filtration. While the components of the MSC are eluted in early fractions as a large molecular weight complex, the MSC-unbound proteins are detected in later fractions according to their individual sizes. As expected, AIMP2 was mainly detected in the early fractions with KARS1, a known component of the MSC[22,39], whereas DX2 was mainly detected in the later fractions (Supplementary Fig. 4e).

**Characterization of the binding mode of DX2 and KRAS.** To elucidate the binding mode of the two proteins, we performed nuclear magnetic resonance (NMR)-based chemical shift perturbation (CSP) analysis using their interaction domains, $^{15}$N-labeled DX2$_{51-251}$[40] in the presence and absence of KRAS4B HVR peptide. DX2 H84, T85, K90, and W120 residues showed strong perturbation while L64 and V83 were weakly perturbed (Fig. 3a), implying the direct binding of KRAS4B HVR to the DX2 GST domain. To validate the NMR results, we generated alanine-substitution mutants of the above residues and examined the mutational effects on the interaction by immunoprecipitation analysis. The results showed that T85, K90, and V92 were significant for the interaction of the two proteins (Supplementary Fig. 5a). Mapping and validation of the perturbed residues suggested that the linearly aligned hydrophobic surface formed between the α-helix bundle and β-sheet of the DX2 GST domain could be a potential cleft for binding of KRAS4B HVR (Fig. 3b and Supplementary Fig. 5b). We also performed $^{1}$H-$^{15}$N transverse relaxation-optimized spectroscopy (TROSY) experiments of $^{15}$N-labeled DX2$_{51-251}$ in the presence and absence of HRAS- or RALA-HVR peptides for comparison. The addition of HRAS- or RALA-HVR peptide induced significant line-broadening of NMR signals, which led to the disappearance of peaks (Supplementary Fig. 5c), implying that these peptides would bind non-specifically to DX2 with intermediate exchange in μs to ms timescale, or induce non-specific aggregation of DX2, not making the specific binding to DX2.

We also conducted a molecular modeling study to build a model that could explain the detailed binding mode for the two proteins, in agreement with the NMR results (see Methods for more details). We performed a protein-protein docking simulation based on the NMR results. We conducted nine molecular dynamics (MD) simulations of length ~1.6 μs from this docked structure. We selected the largest binding interface observed trajectory as the main trajectory among each of the three runs at 100, ~390, and 500 ns time points (Supplementary Fig. 5d). Hence, the main trajectory of the DX2-KRAS4B complex for ~600 ns was obtained. Because the KRAS4B HVR domain is highly flexible, we constrained the distance between KRAS4B HVR and DX2 using a restraint force to shorten the calculation time. This could increase the probability of KRAS4B HVR interaction with the DX2 GST domain. We found that KRAS4B HVR anchored to the site near H86 of DX2 within 10 ns, and maintained an averaged distance of ~11 Å of Cα atoms between KRAS4B I187 and DX2 H86 during the entire simulation time (Supplementary Fig. 5f). For the first 400 ns, the KRAS4B G domain shifted to form a complementary interaction with the DX2 GST domain and was well maintained for the remaining 200 ns (Supplementary Fig. 5e, g). Although the restraint force was removed at 500 ns, the binding of KRAS4B HVR and DX2 was stably maintained for the last 100 ns (Supplementary Movie 1). A representative snapshot at 533.7 ns revealed that

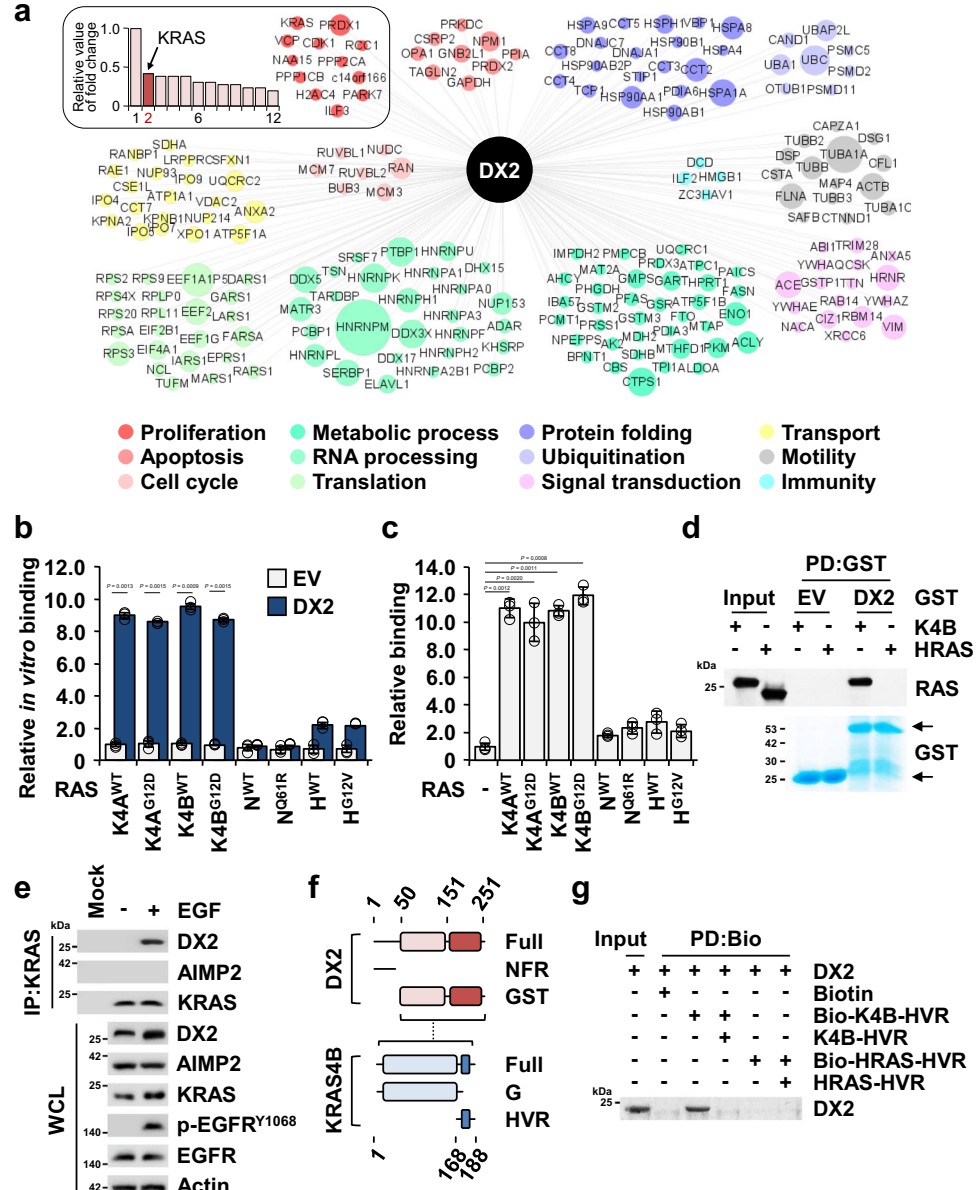

**Fig. 2 Binding of DX2 to KRAS. a** Top 200 proteins identified as the potential binding proteins of DX2 by LC-MS analysis. The relative fold change values of 12 proteins in proliferative ontology were represented as the bar graph (left upper box). Among 12 identified proteins, PRDX1 was detected at the highest fold change and the fold changes of other proteins were divided by that of PRDX1. The relative values of the other eleven proteins were graded, taking the value of PRDX1 as 1. Diameter of circles denotes the relative fold changes of a total of 200 identified proteins. **b, c** Determination of specific binding of DX2 to KRAS. DX2 was mixed with different nanoluciferase-RAS isoforms. Amounts of the indicated RAS isoforms co-precipitated with DX2 were determined by luciferase activity and are shown as a bar graph (**b**). CCD18CO cells expressing Strep-tagged DX2 and the indicated nanoluciferase-RAS isoforms were precipitated using a strep-tactin column. Amounts of the indicated nanoluciferase-RAS isoforms co-precipitated with DX2 were measured and shown as above (**c**). All the experiments were independently repeated thrice and error bars denote S.D. Data are presented as mean values ± S.D. *P* value is from the two-sided *t* test. **d** In vitro pull-down assay showing direct binding between DX2 and KRAS4B. Co-precipitated KRAS4B or HRAS with GST-DX2 was detected by SDS-PAGE and immunoblotting using an anti-pan RAS antibody. GST proteins were detected by Coomassie staining. Results are representative of at least three independent experiments. **e** Endogenous KRAS in DLD-1 cells treated with EGF for 30 min was precipitated with the anti-KRAS antibody. p-EGFR was used as a positive control for EGF signaling. Results are representative of at least three independent experiments. **f** Schematic diagram presenting the binding region between DX2 and KRAS4B. Binding region is depicted as a dashed line (Supplementary Fig. 4a). **g** In vitro pull-down assay showing direct interaction between DX2 protein and KRAS4B, but not with HRAS, HVR peptide. Biotin-conjugated and native peptide were serially mixed with DX2 proteins for checking the binding specificity. Results are representative of at least three independent experiments. EV, PD, IP, and WCL represent empty vector, pull-down, immunoprecipitation, and whole cell lysate, respectively. Source data are provided as a Source data file.

the two interfaces of the DX2-KRAS4B complex are critical binding spots (Fig. 3c and Supplementary Fig. 5h). The distances of the key residue pairs (Supplementary Table 3) obtained from the representative snapshot were calculated for the entire simulation time. We observed that most of the distances gradually

decreased during 500 ns and remained stable in the last 100 ns (Supplementary Fig. 5i, j). To compare the binding mode of DX2 to KRAS4B with or without GTP, KRAS4B structure from the representative snapshot at 533.7 ns was also superimposed with the initial structure for GTP-bound conformation. The difference

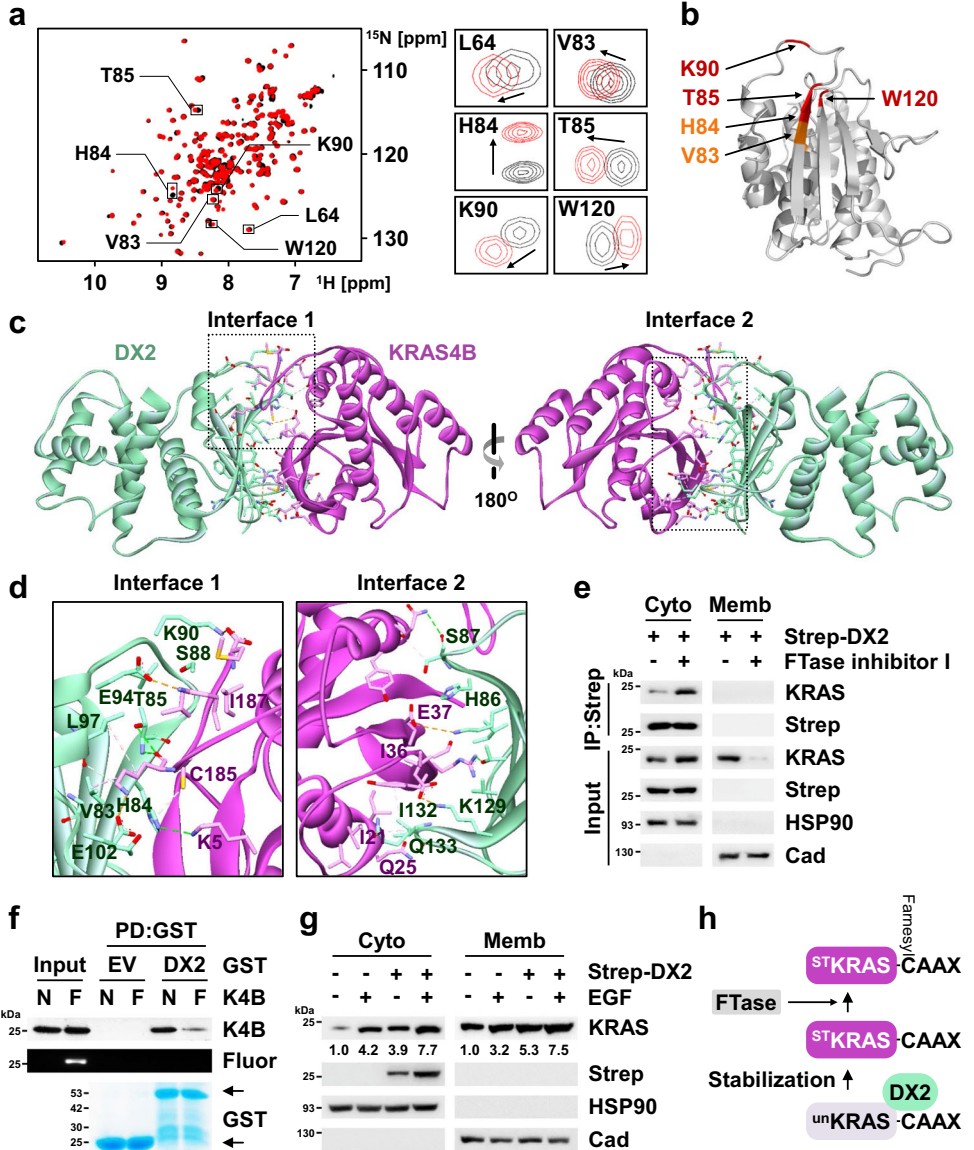

**Fig. 3 Analysis of the binding mode of action between KRAS and DX2. a, b** Chemical shift perturbation of $^{15}$N-labeled DX2$_{51-251}$(C136S, C222S) by the binding to KRAS4B HVR peptide. Superposition of the 2-dimensional (2D) $^{1}$H-$^{15}$N TROSY spectra of free DX2 in the presence (red) and absence (black) of HVR peptide. Strongly affected residues are depicted (**a**, left) and the indicated signals are enlarged and presented (**a**, right). The residues showing strong (Δδ > 0.02 ppm) and medium (0.01 < CSP ≤ 0.02 ppm) perturbation are shown in red and orange, respectively, on the surface of the DX2 GST domain (**b**). **c** Result of 600 ns molecular dynamics (MD) simulation to determine the binding surface of DX2 and KRAS4B. The two significant interfaces are highlighted by the dashed box. **d** Significant residues for binding between DX2 and KRAS4B were analyzed by immunoprecipitation using alanine mutants (Supplementary Fig. 5n, o) and presented in the enlarged image of each interface. **e, f** Validation of the effect of KRAS4B farnesylation on its binding with DX2. H460 cells expressing Strep-DX2 were treated FTase inhibitor I and separated into cytosol and membrane fraction. HSP90 and cadherin (Cad) were used as loading markers for cytosol (Cyto) and membrane (Memb), respectively (**e**). In vitro pull-down assay showing the binding of DX2 to farnesylated KRAS4B. In vitro farnestlyated KRAS4B proteins were mixed with the purified GST-DX2. Binding of the two proteins and farnesylation were determined by immunoblotting and fluorescent scans, respectively. N and F indicate native and farnesylation, respectively (**f**). Results are representative of at least three independent experiments. **g** H460 cells introducing Strep-DX2 were treated with EGF and separated into Cyto and Memb fractions. Relative levels of KRAS were presented below the KRAS blots. Results are representative of at least three independent experiments. **h** Schematic presentation for binding between DX2 and KRAS. $^{un}$KRAS and $^{ST}$KRAS mean unstable and stable KRAS, respectively. Source data are provided as a Source data file.

in Cα-RMSD between KRAS4B with and without GTP is about 1.7 Å (Supplementary Fig. 5k). As shown in Supplementary Fig. 5g, the KRAS G domain maintained the same conformation over the entire simulation time.

We then generated alanine-substitution mutants at the residues predicted to be crucial for binding by contact surface area (Supplementary Fig. 5l, m) and examined their effects on the interactions by immunoprecipitation. The following residues of

the two proteins were selected from the mutation studies: DX2 V83, H84, T85, S88, K90, E94, L97, and E102, and KRAS4B K5, C185, and I187 for interface 1; DX2 H86, S87, K129, I132, and Q133, and KRAS4B I21, Q25, I36, and E37 for interface 2 (Fig. 3d and Supplementary Fig. 5n, o). Interface 1 was identified as the binding surface for the interaction of KRAS4B HVR, C185 and I187, with the hydrophobic cleft of the DX2 GST domain, including H84, T85, S88, K90, and E94 (Fig. 3d, left). These

results are consistent with those of the NMR-based CSP analysis, suggesting the significance of KRAS4B HVR in the interaction with DX2. Interface 2 consists of a pocket surrounded by H86, S87, K129, I132, and Q133 of the DX2 GST domain that interacts with the KRAS4B G domain (Fig. 3d, right). To further validate the significance of these residues in the interactions, the selected alanine mutants were compared with the DX2 WT for their effects on the cellular levels of KRAS4B and observed that all of the tested mutants showed a reduced ability to increase KRAS4B levels (Supplementary Fig. 5p). We also introduced the selected alanine mutants of KRAS4B with DX2 and compared whether their cellular levels were increased by DX2. DX2 increased the cellular level of KRAS4B WT, but not in other mutants (Supplementary Fig. 5q). We then tested the effect of these mutants on cell viability. In contrast to KRAS4B WT, none of the tested mutants increased cell viability (Supplementary Fig. 5r, s). The results of NMR, MD, and mutational analyses collectively suggest the significance of KRAS4B HVR for specific interactions with DX2.

KRAS4B is translocated from the cytosol to the plasma membrane via the Ras converting CAAX endopeptidase 1 (RCE1)-mediated cleavage after farnesylation at HVR C185[38,41]. Because DX2 binds to KRAS4B in the cytosol (Fig. 3e and Supplementary Fig. 3e), and farnesylation and DX2-binding sites of KRAS4B overlap (Fig. 3d), we examined whether pre-farnesylated KRAS4B would bind to DX2 before its HVR is cleaved by RCE1. We observed that the binding of KRAS4B and DX2 was not affected by RCE1 knockdown (Supplementary Fig. 6a). We then checked whether the binding of the two proteins was affected by farnesylation using FTase inhibitor I, a specific inhibitor of RAS farnesylayion[42]. The binding of endogenous KRAS to DX2 was increased by FTase inhibitor treatment (Fig. 3e), and similar results were obtained with ectopically introduced KRAS4A and 4B (Supplementary Fig. 6b). Conversely, we conducted an in vitro pull-down assay using native and farnesylated KRAS with GST-DX2 and found that the farnesylated KRAS4B and KRAS4A showed weaker interactions with DX2 compared to their native counterparts (Fig. 3f and Supplementary Fig. 6c). To determine whether the translocation of KRAS to the plasma membrane was inhibited by DX2, H460 cells with or without Strep-DX2 expression were treated with EGF and KRAS levels were determined in the cytosol and membrane. DX2 increased the KRAS level in the cytosol as well as in the membrane fraction (Fig. 3g), implying that DX2 binding to pre-farnesylated KRAS would have no negative effect on the membrane translocation of KRAS. We also observed consistent results in cell lines expressing either KRAS4A or 4B (Supplementary Fig. 6d). Taken together, DX2 appears to bind and stabilize pre-farnesylated KRAS4B, and subsequently dissociates from KRAS4B upon farnesylation for translocation to the plasma membrane (Fig. 3h).

**DX2 inhibits Smurf2-mediated ubiquitination of KRAS.** We then searched for the E3 ligase responsible for the ubiquitination of KRAS, which is involved in its DX2-dependent stabilization from candidate E3 ligases: SMAD ubiquitination regulatory factor 2 (Smurf2), β-transducing repeat-containing protein (βTRCP), ring-finger protein 40 (RNF40), RING-finger and CHY-zinc-finger domain-containing 1 (RCHY1), and TNF receptor-associated factor (TRAF)-6[43]. Among these, only Smurf2, βTRCP, and RNF40 co-precipitated with GFP-KRAS4B in 293T cells (Supplementary Fig. 7a), while Smurf2 and βTRCP were shown to reduce the cellular levels of KRAS4B (Supplementary Fig. 7b). Since a previous report demonstrated that Smurf2 increases KRAS levels via degradation of βTRCP, an E3

ligase for RAS proteins[44], we examined how these two E3 ligases would affect the endogenous level of KRAS4B in H460 cells. Each of Smurf2 and βTRCP decreased the level of KRAS4B, and co-expression of the two E3 ligases showed stronger suppressive potency (Supplementary Fig. 7c). At this moment, it is not clear what causes the different results of Smurf2 toward KRAS, and further detailed investigation is needed. In this study, we decided to focus on Smurf2 as the E3 ligase for KRAS for further investigation. Exogenous introduction of Smurf2 WT, but not the catalytically inactive (CA) mutants[45], reduced KRAS levels whereas suppression of endogenous Smurf2 increased the KRAS levels (Fig. 4a). Moreover, both KRAS4A and 4B were ubiquitinated by ectopic expression of Smurf2 WT (Supplementary Fig. 8a), but not by the CA mutant (Fig. 4b). We tested whether Smurf2 directly delivered ubiquitin to KRAS4B using an in vitro ubiquitination assay. The conjugation of ubiquitin to KRAS4B was increased in the presence of all the required components: UBE1 (E1), UbcH5c/UBE2D3 (E2), and Smurf2 (E3) (Fig. 4c). The direct binding of Smurf2 to KRAS4B was confirmed by an in vitro pull-down assay using purified proteins (Supplementary Fig. 8b), and the regions responsible for binding of the two proteins were also determined by binding assays using Smurf2 and KRAS4B fragments (Supplementary Fig. 8c–e). The HECT domain responsible for the delivery of ubiquitin[46] was shown to bind to the KRAS4B G domain, suggesting that Smurf2 could function as an E3 ligase for ubiquitination of KRAS4B. Since the G domains of RAS proteins are very similar[2], we checked whether Smurf2 could function as an E3 ligase against other RAS proteins. Smurf2 bound to HRAS as well as to KRAS4B (Supplementary Fig. 8b) and directly ubiquitinated HRAS (Supplementary Fig. 8f), suggesting the possibility of Smurf2 as an E3 ligase to other RAS isoforms beyond KRAS.

We then investigated the effect of DX2 on Smurf2-mediated ubiquitination of KRAS4B. The introduction of DX2 decreased in vitro ubiquitination of KRAS4B, but not HRAS, in a dose-dependent manner (Fig. 4c and Supplementary Fig. 8f), and addition of DX2, not AIMP2, inhibited the direct interaction of Smurf2 and KRAS4B (Fig. 4e and Supplementary Fig. 7g). We also confirmed that DX2 could inhibit the cellular binding of KRAS4A or 4B to Smurf2 by co-immunoprecipitation (Supplementary Fig. 8h). Ubiquitination of KRAS4B in the cytosol was enhanced by Smurf2, but not by co-expression of DX2, further corroborating the importance of DX2 for KRAS4B stabilization (Fig. 4f). DX2 showed no inhibitory effect on Smurf2-mediated ubiquitination of HRAS (Supplementary Fig. 8f), emphasizing the functional specificity of DX2 for KRAS stabilization. To determine whether the competitive relationship between DX2 and Smurf2 on the cellular stability of KRAS would be physiologically meaningful, the effects of EGF signaling on the binding of DX2 and Smurf2 to KRAS were determined. Upon EGF signaling, increased binding of DX2 to KRAS resulted in reduced binding of Smurf2 to KRAS (Fig. 4g, left, and Supplementary Fig. 9a), implying a dynamic relationship between the two factors for the control of KRAS stability. p14ARF and HSP70 have been previously reported as DX2-binding proteins[27,28]. Thus, we checked how DX2 would encounter these proteins upon EGF signaling over time. In response to EGF signaling, DX2 firstly bound to HSP70 (in 5 min) and then to KRAS (in 10 min) in the cytosol. Binding of DX2 to p14ARF in the nucleus was also observed at 10 min after EGF signaling (Fig. 4g, right, and Supplementary Fig. 9b). Combined with previous results, DX2, protected by HSP70[27], appears to promote cancer cell proliferation via the stabilization of oncogenic factor, KRAS, and via the inhibition of tumor suppressor p14ARF.

To determine the ubiquitination residues of KRAS4B, which is specifically affected by DX2, we used mass spectrometry to

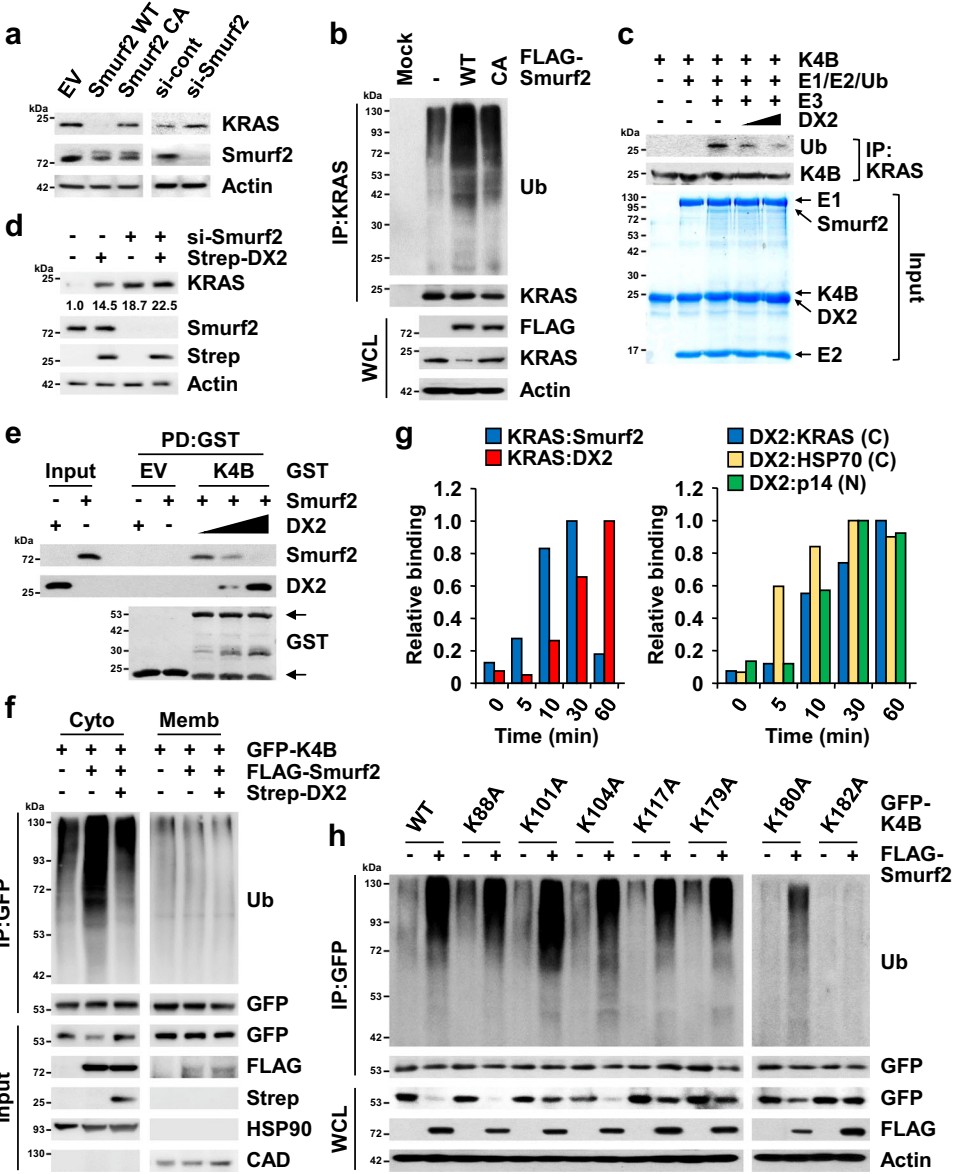

**Fig. 4 Competitive blocking of Smurf2-mediated ubiquitination of KRAS by DX2. a** Smurf2-mediated levels of KRAS. Endogenous levels of KRAS from H460 cells expressing Smurf2 wild type (WT), catalytic-inactive mutant (CA), and si-Smurf2 were determined by immunoblotting. si means siRNA. **b** Smurf2-mediated ubiquitination of KRAS. The cells described in (**a**) were treated with MG-132 and the ubiquitinated amounts of KRAS were analyzed by ubiquitination assay using the anti-ubiquitin (Ub) antibody. **c** In vitro ubiquitination assay showing the effect of DX2 on Smurf2-mediated ubiquitination of KRAS4B. UBE1, UbcH5c/UBE2D3, Smurf2, and KRAS4B were used as E1, E2, E3, and substrate for the reaction, respectively. Proteins were confirmed by Coomassie staining in the input panel (bottom). Ubiquitination of KRAS4B was detected by immunoblotting as above (upper). **d** H460 cells expressing si-Smurf2 and Strep-DX2 were subjected to immunoblotting. Quantitative levels of KRAS were depicted below its blot. **e** In vitro pull-down assay showing the effect of DX2 on the binding of Smurf2 to KRAS4B. Smurf2, GST-KRAS4B, and DX2 proteins were mixed, and co-precipitation of DX2 and Smurf2 with KRAS4B was monitored. **f** Inhibitory effect of DX2 on Smurf2-mediated ubiquitination of KRAS4B. 293T cells expressing GFP-KRAS4B, FLAG-Smurf2, and Strep-DX2 were treated with MG-132 and separated into cytosol and membrane fractions. **g** Analysis of binding kinetics for DX2 or Smurf2 to KRAS (left), and KRAS, HSP70 or p14ARF to DX2 (right) upon EGF signal. All the interactions analyzed by immunoprecipitation (Supplementary Fig. 9a, b) were quantified and presented as graphs. C and N indicate cytosol and nucleus, respectively. The maximum blot intensities obtained from immunoprecipitation of each protein pair was taken as 1, and the other blot intensities were divided by the maximum intensity values and presented as the relative values. **h** Ubiquitination assay validating the Smurf2-dependent ubiquitination sites of KRAS4B. H460 cells expressing GFP-KRAS4B mutants and FLAG-Smurf2 were treated with MG-132 and subjected to the ubiquitination assay. Source data are provided as a Source data file. **a–f**, **h**, Results are representative of at least three independent experiments.

compare the ubiquitination of residues in the presence or absence of DX2 and found that ubiquitination of K179, K180, and K182 in the HVR of KRAS4B was significantly decreased upon ectopic expression of DX2 (Supplementary Fig. 10a). To check whether these residues are ubiquitinated by Smurf2, alanine-substitution mutants at the KRAS4B lysine residues were prepared.

Ubiquitination of the K182A mutant was not affected by the introduction of Smurf2 (Fig. 4h). Interestingly, the K180A mutant also showed much lower ubiquitination compared to the other mutants, implying that the peptide region spanning K180 and K182 residues is a specific site for Smurf2-dependent ubiquitination. Since DX2 binds and stabilizes KRAS4A, we also monitored

Smurf2-dependent ubiquitination of KRAS4A and searched ubiquitination sites using alanine mutants of lysine in KRAS4A HVR. Smurf2 ubiquitinated KRAS4A, similar to KRAS4B, and ubiquitination of K182A mutant was not increased compared to WT (Supplementary Fig. 10b), implying that lysine in HVR is critical for the stabilization of KRAS by blocking Smurf2-mediated ubiquitination.

**Identification and characterization of DX2-KRAS inhibitor**. The positive correlation of DX2 and KRAS levels in cancer cells and patients suggests that the interaction interface may provide a target for suppressing KRAS-driven tumorigenesis. To test this possibility, we set up a nanoluciferase-based complementation assay[47] with DX2 and KRAS4B (Supplementary Fig. 11a) and screened for chemicals that could specifically inhibit the interaction of the two proteins. The primary screening against the interaction of the two factors was designed to identify the compounds that showed over 70% inhibition at 3 μM from the structural diversity chemical library consisting of in-house (1,697) and commercially available compounds (1,102), and the interaction of the protein kinase cAMP-activated catalytic subunit alpha (PRKACA) with the protein kinase cAMP-dependent type II regulatory subunit alpha (PRKAR2A) served as a negative control[47] (Supplementary Fig. 11b). From the 6 hit compounds, we selected the 2-phenylthiophene analogue, BC-DXI-32982 (DXI hereafter, Fig. 5a and Supplementary Fig. 11b), for further studies because it was the most potent inhibitor of the interaction between DX2 and KRAS4B with a half-maximal inhibitory concentration (IC$_{50}$) of 0.18 μM, showing little effect on the PRKACA-PRKAR2A interaction (Fig. 5b). DXI showed inhibitory efficacy on the binding of endogenous DX2 to KRAS (Fig. 5c), but no effect on p14ARF in co-immunoprecipitation (Supplementary Fig. 11c). We also confirmed that BI-2852, the reported KRAS inhibitor[48], did not affect the binding of the two proteins (Supplementary Fig. 11d). DXI also efficiently inhibited the direct binding of the two proteins in the in vitro pull-down assays (Fig. 5d). These results suggest that DXI is a specific inhibitor of the DX2-KRAS interaction. Addition of the compound to H460 cells dose-dependently reduced endogenous KRAS levels and the activities of downstream effectors, phosphorylated extracellular signal-regulated kinase (p-ERK) and phosphorylated protein kinase B (p-Akt) (Fig. 5e). In contrast, the cellular levels of several components of the multi-tRNA synthetase complex (MSC) were unaffected by DXI treatment (Supplementary Fig. 11e). DXI increased ubiquitination of KRAS in a dose-dependent manner, with little effect on the ubiquitination of global cellular proteins (Fig. 5f and Supplementary Fig. 11f). These results support the specificity of DXI action for the DX2-KRAS pair.

We compared the half-maximal effective concentration (EC$_{50}$) values of DXI in suppressing cell viability in various lung, colorectal, and pancreatic cell lines expressing different levels of DX2 and found that the cell lines with high levels of DX2 generally showed higher sensitivity to DXI (Fig. 5g and Supplementary Fig. 12a). DXI also showed no effect on the viability of untransformed MEF cells (Supplementary Fig. 12b). To understand the anti-cancer activity of DXI via KRAS, we monitored the effects of DXI on the activities of ERK, caspase 3, and E-cadherin, which are known to be associated with KRAS activity in diverse cancer cells, such as H460 and A549 (with KRAS mutation), and HCC1588 and H1650 (with KRAS WT) cell lines[36,49]. DXI treatment suppressed phosphorylation of ERK, a KRAS downstream effector, in all tested cell lines regardless of KRAS mutation status (Supplementary Fig. 12c). However, the compound increased the cleavage of caspase 3 and E-cadherin levels only in HCC1588 and H1650 (with KRAS WT), but not in

KRAS mutant cell lines (Supplementary Fig. 12c, d), as previously reported[49]. These results suggest that DXI exert its cytotoxic activity by suppressing ERK via KRAS.

Before checking the in vivo anti-tumor effect, we performed pharmacokinetic profiling of DXI in mice (Supplementary Note 1). Following a single intravenous (iv) and oral dosing at 5 and 10 mg/kg each, the plasma pharmacokinetic data showed a high systemic clearance (CL$_P$ = 4.4 ± 1.8 L/h/kg) and a poor oral bioavailability (F = 0.7 ± 0.3%). Even though DXI was rapidly cleared from plasma (t$_{1/2}$ = 0.5 ± 0.2 h), the plasma concentration (127.6 ng/mL = 0.26 μM) at 1 h after iv dosing was still higher than its EC$_{50}$ (0.18 μM) against H460 cells. Since cumulative dosing of DXI could maintain an effective concentration, we chose 1 and 5 mg/kg for in vivo efficacy experiments. The in vivo tumor-suppressive efficacy of DXI was tested in a mouse xenograft assay using the H460 cell line. Treatment with the compound 5 times at 1 and 5 mg/kg in 12 days reduced the tumor size and weight in a dose-dependent manner, with no effect on body weight (Fig. 5h and Supplementary Fig. 12e–g). The KRAS levels in tumor tissues excised from the DXI-treated group were significantly lower than those in the untreated group (Supplementary Fig. 12h), showing the correlation of its anti-tumor efficacy with the decrease in KRAS levels. We also evaluated the in vivo anti-tumor efficacy of DXI using HCC1588 (KRAS WT) and A549 (KRAS mutant) cell lines. The compound showed similar tumor-suppressive effects on these two cell lines (Fig. 5i and Supplementary Fig. 12i–k), as shown by its effects on cell viability, suggesting that DXI could inhibit cell growth regardless of KRAS mutation. To exclude the possibility of SABV (sex as a biological variable), we also compared the in vivo efficacy of DXI in male and female mice growing tumors of H460 cells, and observed similar efficacy regardless of sex (Supplementary Fig. 12m–p). We analyzed the DXI levels in the tumor tissues obtained from the above studies, determined as 23571 (HCC1588), 8044 (A549), 11242 (H460, male), and 7917 (H460, female) ng/g, which were sufficient for its efficacy (Supplementary Fig. 12l, q, and Supplementary Note 1).

**Profiling of BC-DXI-32982**. We further investigated the off-target effects and in vivo toxicity of DXI. The off-target effect of the compound was characterized using SafetyScreen44-Panlabs (Supplementary Note 1). Among the 44 panel assays, significant responses above 50% at 10 μM were noted in five primary assays including acetyl cholinesterase (66%), β$_2$-adrenergic receptor (55%), L-type calcium channel (63%), cholecystokinin CCK$_1$ (54%), and site 2 sodium channel (75%). Since these five targets are not directly related to ubiquitination-mediated degradation of KRAS, the anti-tumor efficacy of DXI appears to mainly result from the inhibition of the DX2-KRAS interaction. We also examined the in vivo toxicity of DXI after 14-day repeated treatment in ICR male mice (Supplementary Note 1). The dose was chosen as 10 and 50 mg/kg once a day, which is 10 times higher than those used for the efficacy test. No mouse death or macroscopic abnormalities were detected, although a slight increase with no statistical significance was observed in liver weight and in aspartate aminotransferase (AST) and alanine aminotransferase (ALT) activities by blood biochemical analysis. Microscopic analysis revealed hypertrophy of hepatocyte in both of the 10 and 50 mg/kg treated groups. Although hepatotoxicity is not decided solely by hypertrophy, careful monitoring of related symptoms may be needed. The values of monocytes and large unstained cells (LUCs) were significantly increased, and infiltration of mixed cells in the lung was observed in the 50 mg/kg treated group. However, the changes were not severe enough to be considered as a result of toxicity.

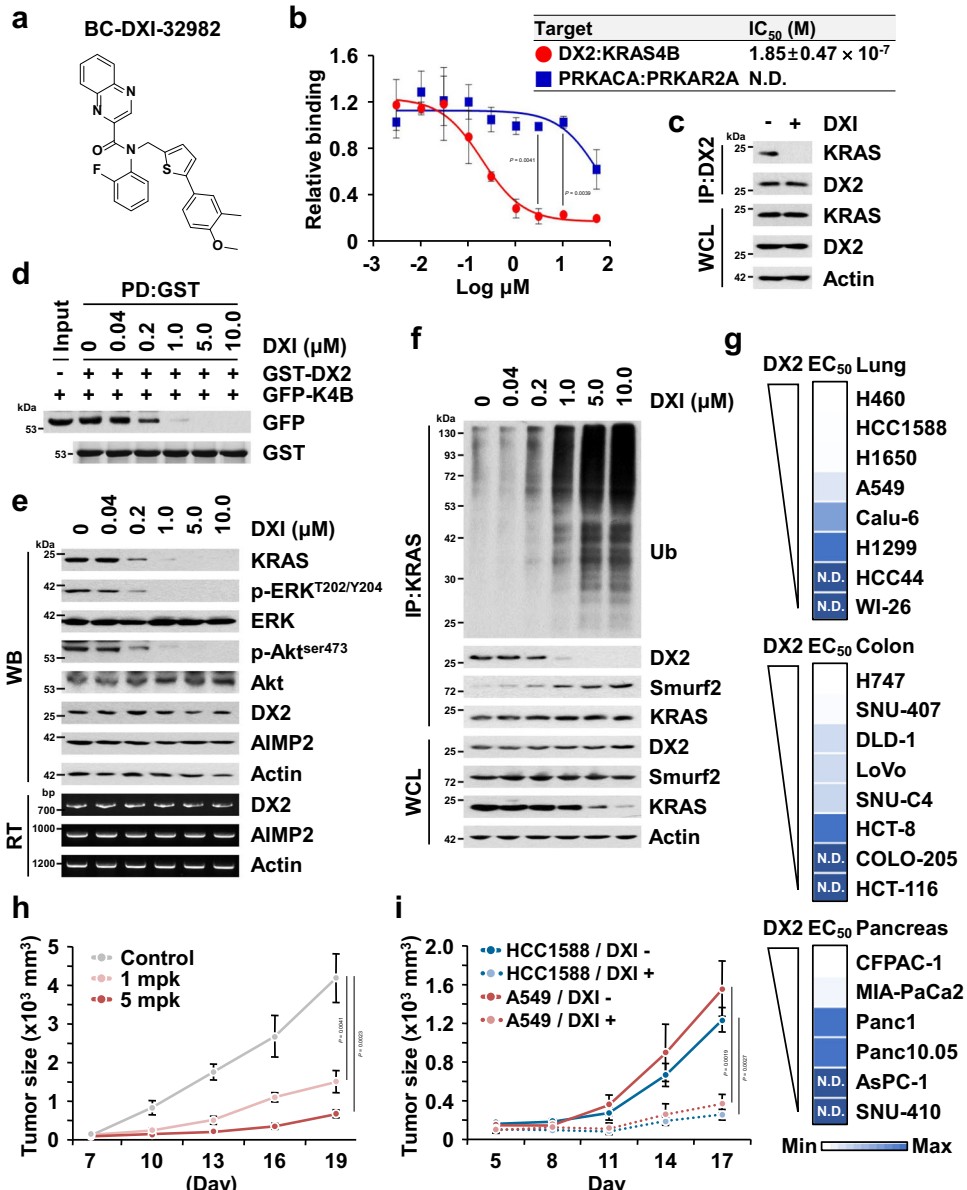

**Fig. 5 Identification and characterization of BC-DXI-32982. a** Structure of BC-DXI-32982 (DXI hereafter). **b** Inhibitory effect of DXI on the binding of DX2 and KRAS4B. Binding pair of PRKACA and PRKAR2A was used for checking the binding specificity. N.D. denotes "not determined" in the tested range of dose. The experiment was independently repeated thrice and error bars denote S.D. Data are presented as mean values ± S.D. *P* value is from the two-sided *t* test. **c** Endogenous immunoprecipitation showing the interaction between DX2 and KRAS in H460 cells treated with DXI. Results are representative of at least three independent experiments. **d** In vitro pull-down assay showing direct inhibition of DXI on the binding of DX2 and KRAS4B. DXI was incubated with the mixture of cellular extracts expressing GFP-KRAS4B and the purified GST-DX2 proteins. Results are representative of at least three independent experiments. **e** Suppression of KRAS level via DXI. H460 cells treated with various concentration of DXI for 18 h were subjected to immunoblotting and RT-PCR. Results are representative of at least three independent experiments. **f** Ubiquitination assay analyzing DXI-mediated ubiquitination of KRAS. Results are representative of at least three independent experiments. **g** Heat map showing EC50 of DXI on lung, colorectal, and pancreatic cancer cell lines expressing different levels of DX2 (Supplementary Fig. 12a). In each type of cancers, the maximum and minimum EC50 values were represented as blue color with the highest and lowest intensity, respectively, and the rests were graded according to their relative values. **h**, **i** In vivo efficacy of DXI on tumor growth. Mice (*n* = 3) xenografted with H460 cells were intraperitoneally injected with DXI (1 or 5 mg/kg) five times in 12 d (**h**). Mice (*n* = 6) xenografted with HCC1588 or A549 cells were intravenously injected with DXI (5 mg/kg) five times during the experiment (**i**). All error bars represent the standard deviation (S.D.). *P* value is from the two-sided *t* test. Source data are provided as a Source data file.

**Mode of action of BC-DXI-32982.** To understand the actual binding mode of DXI, we synthesized two biotin-conjugated DXI compounds (Biotin-DXI #1 and #2) (Supplementary Figs. 14, 15, and Supplementary Note 2) and evaluated their effect on KRAS levels. We then selected Biotin-DXI #2 for further study because it reduced KRAS level more effectively than Biotin-DXI #1 (Supplementary Fig. 13a). We also observed that Biotin-DXI #2

efficiently inhibited the DX2 and KRAS4B interactions in in vitro pull-down and immunoprecipitation assays (Supplementary Fig. 13b, c). We tested the binding of Biotin-DXI #2 to DX2 or KRAS4B via an in vitro pull-down assay using purified proteins and found that the compound bound to DX2 but not to KRAS4B (Fig. 6a). To further confirm the specificity of the compound to DX2, we monitored the competition between the original and

biotin-conjugated compounds to DX2. We confirmed that Biotin-DXI #2 binding to DX2 was inhibited by the addition of DX1 (Fig. 6b), indicating that it would bind to DX2 with the same binding mode as DXI. The Kd value of DXI to DX2 was estimated as 480 nM (Supplementary Fig. 13d). Binding assays using different fragments of DX2, such as the NFR, GST domain, and GST-C sub-domain, revealed NFR and GST-N as potential binding regions for the compound (Supplementary Fig. 13e). Since the GST domain is present in both DX2 and AIMP2[27], we checked whether DXI could also bind to AIMP2. Although DXI showed binding to the purified AIMP2 in an in vitro pull-down assay, it did not bind to cellular AIMP2 in an immunoprecipitation assay (Supplementary Fig. 13f, g). Moreover, DXI did not affect the cellular binding of AIMP2 with KARS1 (a component of the MSC) and p53 (an interactor in the nucleus)[19,39] (Supplementary Fig. 13h, i), confirming the specificity of DXI action to the DX2-KRAS interaction. Based on these results, DXI was more likely to affect the tumor-promoting interactions of DX2 and KRAS.

Next, we performed molecular docking and ~400 ns MD simulation. To improve the likelihood of detecting the binding event, the MD simulation was performed with an upper-wall restraint force to favor localization of the ligand close to the surface of the DX2 GST-N domain. The trajectory showed initial random localization of the ligand followed by a hydrophobic fit into the β-sheet of the GST-N domain for the first ~150 ns. In the last 250 ns, the ligand maintained stable interactions with five key interacting residues, Y47, G48, V54, I119, and K129, located in a pocket comprising a β-sheet surface of the DX2 GST domain (Supplementary Movie 2). We used the snapshot at 371.2 ns, which represents the highly populated conformation of the ligand during the last 250 ns that has the lowest non-bond energy between protein and ligand (Fig. 6c, upper). The binding of DXI with DX2 showed that the phenylthiophene core and fluorophenyl moiety interacted with the lysine-rich region on the GST domain of DX2 via π-cation interactions and the quinoxaline ring interacted with the surrounding tyrosine by π-π stacking. To validate the pose stability of the representative snapshot, we conducted three replica simulations without restrain force starting from the snapshot at 371.2 ns for each 100 ns. The values of Cα-root mean square deviation (RMSD), ligand RMSD, and interaction energy for the three replica systems were stably maintained for the entire simulation time (Supplementary Fig. 13j, k). These results prove the binding pose stability of the proposed model. To validate the predicted binding model, we introduced alanine substitutions at 15 residues of DX2 that were proposed to be involved in the interaction with the compound and evaluated the effects of mutations on compound binding. The alanine mutations at Y47, G48, V54, I119, and K129 reduced more than 50% of the compound's binding ability to DX2 and the suppressive effect on KRAS level and the DX2-KRAS4B interaction (Fig. 6d and Supplementary Fig. 13n–p), suggesting that these five residues could be crucial for the interaction with the compound (Fig. 6c, d). We also introduced these mutants into A549 cells and tested the effect of DXI on DX2-mediated cell viability and transformation. Consistent with the results described above, the effects of chemicals on cell proliferation and transformation were not observed in the cells expressing the mutants (Supplementary Figs. 13q, 6e). To confirm the impact of the alanine mutations, the MD simulation of Y47A, one of the effective mutants, was performed for ~380 ns. From the results of RMSDs and interaction energy, we observed an apparent destabilization of the complex with the Y47A mutant compared to WT (Supplementary Fig. 13l, m). The MD snapshots of DX2 with the KRAS4B (533.7 ns) and DXI (371.2 ns) complexes were superimposed on the DX2 structure (Fig. 6c, bottom). When DXI was bound to DX2, a region of KRAS4B harboring I36 and E37 overlapped with the quinoxaline

moiety of DXI, suggesting that binding of DXI to the β-sheet of the DX2 GST domain interferes with DX2-KRAS4B complex formation. These results indicate that DXI competitively interferes with the interaction between DX2 and KRAS through its binding to the DX2 GST domain, leading to the suppression of KRAS-driven cancer cell growth.

## Discussion

KRAS activity is often enhanced in cancers via hyperactivating mutations[4,5] and gene amplification[50–53]. However, little is known about how the cellular levels of KRAS are regulated at the protein level. Although it was previously reported that RAS proteins can be controlled by protein kinase Cδ (PKCδ)[54], glycogen synthase kinase 3β (GSK3β)[55], leucine zipper-like transcriptional regulator 1 (LZTR1)[56], and Rab5 guanine nucleotide exchange factor (Rabex-5)[57], these processes do not distinguish KRAS from other isoforms, and the exact mechanism is not yet fully understood. Here, we report a mechanism whereby KRAS stability is specifically controlled by another oncogenic factor, DX2. We also identified Smurf2 as one of the E3 ligases of KRAS and found that DX2 prevents KRAS from Smurf2-mediated degradation in the cytosol. The specificity of DX2 to KRAS results from its binding to KRAS HVR. Because the disordered structure of KRAS HVR provided technical difficulties for structural analysis, we conducted MD simulations to deduce the interaction mode and confirmed the structural prediction by mutation studies. Although DX2 does not distinguish between KRAS WT and the known oncogenic mutants, the cancer-specific expression of DX2[24,25] suggests a pathological implication of the interaction of the two factors for tumor promotion.

Our MD simulation and mutagenesis analysis revealed the significance of C185 and I187 in KRAS4B HVR for binding with DX2 (Fig. 3d). Intriguingly, these residues are critical for membrane trafficking of KRAS4B via farnesylation and subsequent cleavage process[38,41], posing the question of how KRAS farnesylation affects the role of DX2 in KRAS stabilization. DX2 binds cytosolic, not membranous, KRAS4B, and the binding of the two proteins was not affected by knockdown of RCE1 protein (Supplementary Figs. 3e, 6a). These results, along with other supporting data, suggests that DX2 contributes to the stabilization of cytosolic KRAS prior to farnesylation and membrane localization (Fig. 3h). MD results also revealed that the residues of the KRAS4B G domain are crucial for DX2 binding. Among the validated critical residues, K5 and I36 were reported to be associated with gastric cancer and noonan syndrome[58,59], although their pathological and mechanistic connections await further investigation. Our interactome analysis identified multiple potential interactions of DX2 in EGF-induced proliferative conditions. For instance, PRDX1, detected at the highest frequency (Fig. 2a), plays a role in protecting cells against oxidative stress by detoxifying peroxides[37]. Although EGF signal dependency was not observed for the interaction between PRDX1 and DX2 (Supplementary Fig. 3d), DX2 may play a cellular role in oxidative stress via PRDX1.

Uncontrolled proliferation and transformation of cancer cells requires the activations of oncogenes and inhibition of tumor suppressors[60]. Upon EGF signaling, DX2 also moves to the nucleus, suppressing the tumor suppressor p14ARF. Considering the multifaceted roles of DX2 in tumorigenesis, targeting DX2 would provide an efficient way to control a broad spectrum of cancers.

For decades, direct control of oncogenic KRAS mutants has been frustrating due to its high affinity to nucleotide substrates and subtle structural differences between the WT and mutant forms[3,4,6]. Although recent advances in covalent inhibitors of

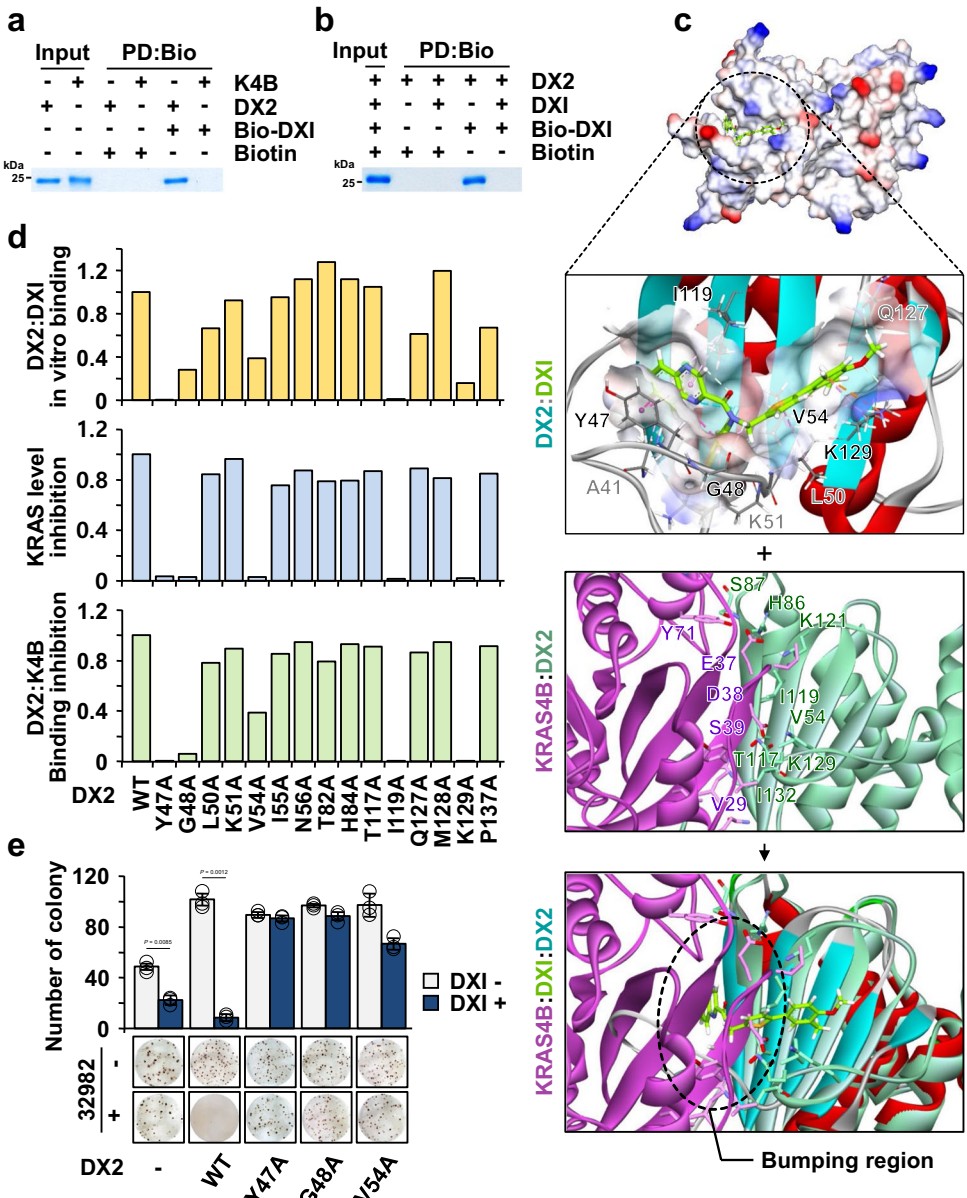

**Fig. 6 Mode of action of BC-DXI-32982. a** In vitro pull-down assay showing the direct binding of DXI with DX2, but not with KRAS4B. Proteins co-precipitated with Biotin-DXI #2 (Bio-DXI) were determined by SDS-PAGE and Coomassie staining. Biotin was used as a negative control. Results are representative of at least three independent experiments. **b** In vitro pull-down assay showing the specific interaction of compound with DX2 using Bio-DXI and DXI. Results are representative of at least three independent experiments. **c** Binding mode of DXI to DX2 as obtained from the MD simulation. The representative structure at 371.2 ns is shown as an electrostatic surface model (first from above). Zoomed-in view of the detailed interactions is presented and binding residues of the compound (green) were depicted as a stick model (second from above). MD snapshots of DX2 and KRAS4B complex at 533.7 ns (third from above). Overlay of above two MD snapshots by superimposing over DX2. The dashed circle denotes the contact region proposed to be responsible for the DXI-mediated inhibition of the interaction of DX2 with KRAS4B (bottom). **d** Graphs showing the critical residues of DX2 for binding with DXI. Results from the in vitro binding of DX2 mutants with DXI; compound-mediated change in KRAS levels and inhibition of DX2 mutants-KRAS4B were quantified (Supplementary Fig. 13n–p). **e** Anchorage-independent colony formation assay validating the significance of the binding between DXI and DX2. Representative images are shown (bottom). The experiment was independently repeated thrice and error bars denote S.D. Data are presented as mean values ± S.D. *P* value is from the two-sided *t* test. Source data are provided as a Source data file.

KRAS have shown promise for mutant-specific drug development[9,61], an alternative approach to deal with diverse KRAS-driven cancers still needs to be devised. While a vast amount of genetic data is available for the cancer-associated KRAS gene, how KRAS is controlled at the post-transcriptional level is not well understood. Here, we show a tumor-promoting stabilization mechanism of KRAS *via* another oncogenic protein, DX2, and propose the interface of the two oncogenic factors as an alternative and effective route to control KRAS-driven cancers.

We validated the efficacy of targeting the DX2-KRAS interaction by demonstrating the suppressive effect of DXI on KRAS-driven tumors that expressed high levels of DX2. We further confirmed that the anti-cancer efficacy of DXI mainly results from its inhibitory activity against the DX2-KRAS interaction, not from the off-target effects (Supplementary Note 1). This work also suggests that DXI is an interesting lead compound with the potential to treat cancers with oncogenic mutations or expression of KRAS and high levels of DX2 (Fig. 5h, i and Supplementary

Fig. 12), even if further optimization is needed for better efficacy and safety.

Notably, diverse cancer cell lines and tissues from patients with cancer showed a close association between KRAS and DX2 levels (Fig. 1g, h), suggesting that patients with high KRAS and DX2 levels could respond more sensitively to the inhibition of the KRAS and DX2 interaction than those with low levels of the two proteins. Interestingly, KRAS oncogenic mutations were frequently found in colorectal cancer cell lines showing high levels of both DX2 and KRAS (Supplementary Table 1), while no clear correlation was observed between KRAS level and mutations in lung and pancreatic cancer cell lines. Therefore, further investigation of the oncogenic potential of high KRAS and DX2 levels would help to validate the significance of DX2-targeting therapeutics against KRAS-driven cancers.

## Methods

**Cell culture and materials**. COLO-205, HCT-8, KM-12, SNU-C4, SW-403, NCI-H747, SNU-407, NCI-H1666, NCI-H1650, Calu-6, NCI-H441, HCC1588, and SNU-410 cell lines were purchased from the Korean Cell Line Bank. CCD18CO, HCT-116, DLD-1, LoVo, WI-26, H460, AsPC-1, Panc10.05, BxPC3, SU.86.86, CFPAC-1, MIA-PaCa2, H69, WI-38, HCC44, H1975, HCC2108, H1299, H1792, H226, A549, HCC827, H520, CaCo2, Panc1, 293T, and CHO-K1 cell lines were obtained from the BioBank of Medicinal Bioconvergence Research Center (Biocon). CCD18CO, WI-26, WI-38, MIA-PaCa2, and 293T cells were cultured in Dulbecco's modified Eagle's medium and other cell lines in the Roswell Park Memorial Institute (RPMI) medium supplemented with 10% fetal bovine serum (FBS) and 1% penicillin/streptomycin in 5% carbon dioxide ($CO_2$) at 37 °C. All plasmids for GFP-tagged RAS proteins (kindly gifted by Dr. Mark R. Philips) were subcloned at the $EcoRI/XhoI$ sites of the pNL1.1 vector (Promega). Human DX2 was cloned at the $EcoRI/XhoI$ sites of the pEXPR-IBA5 vector. Point mutagenesis of DX2 and RAS was performed using Quik-ChangeII (Promega) following the manufacturer's instructions. Cytosol, membrane, and nuclear fractions were separated using a ProteoExtract kit (Calbiochem) following the manufacturer's instructions. Purified KRAS4B and HRAS proteins were purchased from Abcam. FTase inhibitor I and siRNAs against RCE1 and KRAS were purchased from Santa Cruz Biotechnology, and siRNA specifically targeting DX2 (UCAGCGCCCCGUAAUC CUGCACGUG)[24] were purchased from Invitrogen. EGF, and doxycycline, cycloheximide, MG-132, and puromycin were purchased from Peprotech and Sigma, respectively. Specific antibodies against AIMP2 and DX2 were purchased from Curebio. FLAG, actin, and pan-RAS antibodies were purchased from Sigma-Aldrich. Strep, RCE1, and p14ARF were purchased from IBA, Novus Biologicals, and Merck, respectively. HRAS and PRDX1 antibodies were purchased from Thermo Fisher Scientific. Antibodies recognizing p-ERK, ERK, p-Akt, Akt, p-EGFR, EGFR, and cleaved caspase-3 were purchased from Cell Signaling Technology and other antibodies were from Santa Cruz Biotechnology. Dilution fold and catalog number using antibodies are as follows; anti-DX2 (1:1000, NMS-02-0012), -AIMP2 (1:1000, NMS-02-0011), -FLAG (1:5000, F3165), -Actin (1:5000, A1978), -pan-RAS (1:1000, MABS195), -Strep (1:10000, 2-1509-001), -RCE1 (1:500, NBP1-59922), -KRAS (1:50, NBP2-33579), -p14ARF (1:500, MAB3782), -HRAS (1:500, 18295-1-AP), -PRDX1 (1:1000, PA3-750), -p-ERK (1:1000, #9101), -ERK (1:1000, #4695), -p-Akt (1:1000, #4060), -Akt (1:1000, #9272), -EGFR (1:1000, #4267), -p-EGFR (1:1000, #3777), -cleaved caspase-3 (1:1000, #9661), -KRAS (1:500, sc-30), -GFP (1:1000, sc-9996), -HSP90 (1:1000, sc-13119), -pan-cadherin (1:500, sc-515872), -Smurf2 (1:500, sc-518164), -ubiquitin (1:500, sc-53509), -HA (1:1000, sc-7392), -myc (1:1000, sc-40), -HSP70 (1:1000, sc-24), -YY1 (1:1000, sc-7341), and -E-cadherin (1:500, sc-8426).

### Molecular docking simulation

*Preparation of systems*. The crystal structure of GTP-bound HRAS (PDB ID: 1NVU, Chain Q)[62] was mutated to the KRAS4B sequence because the RAS isoforms showed no significant structural differences. The mutated structure was constructed by the "Build" and "Edit Protein" tools in Discovery Studio (DS) 2018 software[63]. For DX2, the crystal structure of AIMP2 GST domain (PDB ID: 5A34)[18] was used and also edited by the "Build" and "Edit Protein" tools in DS 2018. The initial structure of the DX2 and KRAS4B complex was generated by ZDOCK protein-protein docking simulation[64] of DS 2018, based on the NMR results considering binding residues H84, T85, H86, S87, S88, K90, N95, E102, K105, W120, and L249 of DX2, and E3, Y4, K5, K16, I21, Q25, H27, I36, E37, D57, E62, T74, G75, E76, and T87 of KRAS4B. The largest binding interface observed model (cluster 13) was selected as the final docking result and this docked structure was subjected to molecular dynamics (MD) simulation. The MD simulation system consisted of 132,339 atoms, including 6,232 atoms for the proteins and ~42,000 water molecules. It was constructed with a cubic water box at the lowest edge distance of 10 Å from the protein. The system was neutralized with 119 $Na^{2+}$ and 129 $Cl^-$ ions, resulting in ~150 mM salt concentration in a box size of ~11 × 11 × 11 nm$^3$ with periodic boundary conditions. The simulation was conducted for ~600 ns. To predict the binding mode of DXI with DX2, molecular

docking simulation of DXI was performed using CDOCKER[65] implemented in DS 2018. The binding site of DX2 was defined as the surface of the β-sheet of the GST N-terminal domain (GST-N, amino acids 51–151). Our previous data, which provided a representative structure obtained from MD simulation including a relaxed conformation of the N-terminal flexible region (NFR), was used as the initial structure of DX2 for the docking of DXI[66]. The highest-scoring binding conformation was selected as the initial structure for MD simulation. The DX2 and DXI complex structures, including 3,362 atoms of DX2 alone and 57 atoms for the ligand, were immersed in a fully solvated cubic box of ~8×8×8 nm$^3$ by ~14,200 water molecules with periodic boundary conditions. The resulting MD system comprised 46,067 atoms. The system was neutralized with 40 $Na^{2+}$ and 47 $Cl^-$ ions, corresponding to a salt concentration of ~150 mM. The simulation was conducted for ~400 ns.

*Details of MD simulations*. All the MD simulations were carried out using GROMACS software (version 2016) with a charmm36 force field, and the Plumed plugin version 2.4[67] was used for the upper-wall restraints. The CHARMM-GUI[68,69] and CHARMM General Force Field (CGenFF) program[70] were used to generate the input topologies and parameters. The total energy of the system was minimized using the steepest descent algorithm to remove possible bad contacts and reach a tolerance value of 1000 kJ/(mol·nm). The minimized system was subjected to position-restrained MD simulations for 25 ps with a 1 fs time step in the NVT ensemble at a constant volume and temperature of 303.15 K. Production runs of the DX2-KRAS4B and DX2-DXI complexes were performed for ~600 ns and ~400 ns, respectively, in the NPT ensemble at a constant temperature of 303.15 K and pressure of 1 bar, which was achieved using the Nosé–Hoover thermostat[71] and Parrinello-Rahman barostat[72], respectively. Both the cut-off values of short-range van der Waals and electrostatic interactions were set to 12 Å, and the long-range electrostatic interactions were calculated using the Particle Mesh Ewald (PME) method[73]. The LINCS algorithm[74] was used to constrain the bonds between the heavy atoms and the corresponding hydrogen atoms by their equilibrium bond lengths. The time step was set to 2 fs for the production runs, and the coordinate data were saved every picosecond. To increase the probability of KRAS4B HVR binding to DX2, an upper-wall restraint force to the distance between HVR (C-alpha atom of T183) and DX2 (C-alpha atom of E141) was applied for the first 500 ns when the distance exceeded the cut-off limit ($d_{up}$) of 30 Å. The remaining ~100 ns of the simulation were conducted without the use of force. For the DX2-DXI complex, to avoid the ligand from escaping from the binding surface, the restraint force was the distance between the center of mass (COM) of the GST-N domain and the COM of the DXI. The force was applied to the system when the distance exceeded the cut-off limit ($d_{up}$) of 12 Å. The harmonic potentials of the upper wall for the DX2-KRAS4B and DX2-DXI systems were set with a force constant of κ = 100 and 200 kJ/mol·nm$^{-2}$, respectively. The force was calculated as following Eq. 1.

$$\text{Bias}_{up} = \begin{cases} 0 & \text{for} \quad d < d_{up} \\ k \cdot (d - d_{up})^2 & \text{for} \quad d \geq d_{up} \end{cases} \quad (1)$$

The structure having the lowest interaction energy during the last 100 ns was selected as representative MD snapshot for DX2 and KRAS4B complex at 533.71 ns and DX2 and DXI complex at 371.2 ns.

**NMR analysis**. $^{15}N$-labeled DX2$_{51-251}$(C136S and C222S) was overexpressed and purified from *Escherichia coli* strain BL21-CodonPlus(DE3)-RIPL in M9 minimal medium enriched with $^{15}NH_4Cl$ as the sole nitrogen source (99% $^{15}N$; Cambridge Isotope Laboratories)[40]. The $^{1}H$-$^{15}N$ TROSY experiments were performed with 0.3 mM $^{15}N$-labeled DX2$_{51-251}$(C136S and C222S) in the presence and absence of 3.0 mM KRAS4B, HRAS (RKLNPPDESGPGCMSCKC) or RALA (KKKRKSLAK-RIRERC) HVR peptide in buffer 20 mM Bis-Tris (pH 6.0), 100 mM NaCl, 100 mM glycine, 1 mM dithiothreitol (DTT) and 1 mM phenylmethylsulfonyl fluoride (PMSF) at 298 K. KRAS4B HVR peptide was kindly provided by Korea Basic Science Institute (KBSI). To avoid pH changes upon the addition of the peptide, we conducted dialysis in the same beaker, using a membrane with a 500 Da cut off, for both samples with and without the peptide. The backbone assignment of $^{13}C,^{15}N$-labeled DX2$_{51-251}$(C136S and C222S) was performed with a series of triple-resonance two- and three-dimensional experiments. Data were processed with NMRpipe[75] and analyzed using CCPN2.1.5[76]. The backbone assignment of DX2$_{51-251}$(C136S and C222S) was deposited in the Biological Magnetic Resonance Bank (BMRB; http://www.bmrb.wisc.edu/) with accession number 27914. The CSP of $^{15}N$ and $^{1}H$ nuclei was analyzed by overlaying the $^{1}H$–$^{15}N$ TROSY spectra of free protein with those of the KRAS4B HVR peptide. The magnitude of the combined $^{1}H$–$^{15}N$ chemical shift differences (Δδ, ppm) was calculated using the equation $\Delta\delta = (\delta H^2 + 0.2 \times \delta N^2)^{1/2}$, where δH and δN were changed to the proton ($^{1}H$) and nitrogen ($^{15}N$) chemical shifts, respectively[77]. All NMR spectra were recorded using an Avance 600 MHz NMR spectrometer equipped with a triple-resonance probe (Bruker, Germany).

**Screening of chemical inhibitors against the interaction between DX2 and KRAS**. DX2 and KRAS4B were cloned into pBiT1.1-N[TK/LgBiT] and pBiT2.1-N[TK/SmBiT], respectively. Plasmids expressing LgBiT-PRKAR2A and SmBiT-

PRKACA were obtained from Promega. CHO-K1 cells co-transfected with LgBiT-DX2 and SmBiT-KRAS4B were seeded into 96-well plate. After 24 h, the cells were incubated with serum-free media containing each compound obtained from the in-house synthesized 1697 chemical diversity library (Dongguk Univ.) and a commercial 1,102 chemical library (Selleckchem) for 4 h. The in-house library contains synthesized chemicals composed of diverse chemotypes, including phenylthiophene, biphenyl[78], benzofuran[79–82], stilbenes[83,84], aryloxyacetamide[85–94], and sulfonamide[27,95]. After incubation, luciferase activity was detected using the nanoluciferase assay system following the manufacturer's protocol (Promega). The 42 compounds inhibiting the interaction of DX2 and KRAS4B over 70% at 3 μM obtained from primary screening were subjected to counter screening for test specificity using the control binding pair of PRKACA and PRKAR2A. Combining the results from primary and counter screening, six compounds were identified as hit compounds and DXI was selected for further study because of its strong potency in the dose-dependency test. Chemical probes of DXI were designed and synthesized based on molecular modeling of the DX2 and KRAS4B complex for further mechanistic studies. Synthetic procedures for DXI and the biotinylated derivatives (Biotin-DXI #1 and #2) are provided in Supplementary Figs. 14, 15, and Supplementary Note 2.

## Mass spectrometry analysis

*Interactome analysis.* H460 cells expressing Strep-EV or -DX2 were treated with EGF for 30 min, and 5 mg of cell lysates ($n = 1$) were subjected to strep-tactin column chromatography. Eluents were digested in-gel digestion using trypsin/Lys-C (Promega). After purification of tryptic peptides using a C18 spin column, the peptide mixture was analyzed using an LTQ-Orbitrap Velos (Thermo Fisher Scientific) connected to an Easy-nano LC II system (Thermo Fisher Scientific) incorporating an autosampler. One-tenth of the peptides were resuspended in 0.1% formic acid and injected into a reversed-phase peptide trap EASY-Column (length 2 cm, internal diameter 100 and 5 μm, 120 A, ReproSil-Pur C18-AQ; Thermo Fisher Scientific) and a reversed-phase analytical EASY-Column (length 10 cm, internal diameter 75 and 3 μm, 120 A, ReproSil-PurC18-AQ; Thermo Fisher Scientific). ESI was subsequently performed using a 30 μm nano-bore stainless steel online emitter (Thermo Fisher Scientific). The total duration of the LC gradient procedure was 2 h. The LTQ-Orbitrap Velos mass analyzer was operated in positive ESI mode using collision-induced dissociation to fragment the HPLC-separated peptides. The data were analyzed with Sequest (XCorr only; Thermo Fisher Scientific; v.27, rev.11) and X! Tandem (The GPM, thegpm.org; v.CYCLONE 2010.12.01.1) using a human database (Uniprot human, release 2014). The Scaffold program (v.4.6.1, Proteome Software) was used to validate MS/MS-based peptide and protein identifications and to process the quantitative analysis. Fold change value (DX2/EV) of the most frequently detected protein was set to "1", and the remaining values were expressed in proportion.

*Identification of the ubiquitination site.* Strep-DX2 was introduced into 293T cells expressing GFP-KRAS4B. Cells treated with MG-132 and KRAS4B proteins were purified by immunoprecipitation with an anti-GFP antibody ($n = 1$). Equivalent amounts of eluted protein were subjected to sodium dodecyl sulfate-polyacrylamide gel electrophoresis (SDS-PAGE) and Coomassie staining. KRAS4B proteins separated by SDS-PAGE were subjected to in-gel digestion with trypsin GOLD (Promega). Analysis of the peptide mixture was performed using LTQ-Orbitrap Velos (Thermo Fisher Scientific) connected to Easy-nano LC II system (Thermo Fisher Scientific) with an incorporated autosampler. Acquired data were analyzed from the data-dependent mode to simultaneously record full-scan mass and collision-induced dissociation (CID) spectra with multistage activation. Mass spectra were searched against the KRAS4B sequence database (Uniprot accession No.: P01116-2) with Proteome Discoverer (version 1.3, Thermo Scientific, Waltham, MA USA) with the SEQUEST search engine. The precursor mass tolerance and fragment mass tolerance were set to 25 ppm and 0.8 Da. For ubiquitinated peptide identification, lysine ubiquitination (+114.04 Da) and methionine oxidation (+15.99 Da) were set as variable modifications and cysteine carbamidomethylation (+57.02 Da) was set as a static modification.

## Generation of inducible DX2 knock-in mice and embryonic fibroblast cells. To generate inducible human DX2 transgenic mice, the tetO-DX2 construct was prepared under the control of a minimal promoter from hCMV fused to the tetO sequence. This construct was subcloned into the ROSA targeting vector (Soriano P's lab). The targeting vector was electroporated into mouse ES cells (E14TG2a), according to a previously reported procedure[96]. Correctly targeted clones were screened by Southern blot analysis and injected into C57BL/6 blastocysts for chimera generation. Germline transmission of the transgene allele was verified using PCR. Pups with human DX2 transgene were genotyped using the primer for the ROSA locus and maintained in a homozygous colony (ROSA^hDX2/hDX2). CAG-rtTA3 transgenic mice (016532, The Jackson Laboratory) were crossed with hDX2 transgenic mice to generate doxycycline-inducible hDX2 mouse colonies (CAG-rtTA3; ROSA^hDX2/+). A doxycycline diet (TD.01306, Harlan Teklad) was provided to the mice *ad libitum* for 1 month to induce human DX2. The sequences of ROSA locus primers were synthesized using the following sequences: These are ROSA1, AAAGTCGCTCTGAGTTGTTAT; ROSA2: GGCGGGCCATTTACCGTAAG; ROSA3: GGAGCGGGAGAAATGGATATG. MEFs were prepared at embryonic day 13, generated from the cross between CAG-rtTA3 and hDX2 transgenic mice.

Doxycycline-inducible MEFs were selected by genotyping and used for further experiments. Doxycycline (1 μg/mL) was added to the culture medium for hDX2 induction.

## Immunohistochemistry. Tissue microarray (TMA) slides (US Biomax, Inc.) were deparaffinized and then rehydrated in different concentrations of ethanol (100, 95, 80, and 70%). Endogenous peroxidases were removed with 0.3% hydrogen peroxide ($H_2O_2$) (Sigma-Aldrich, St. Louis, USA) in phosphate-buffered saline (PBS) for 10 min, and antigen retrieval was performed using 10 mM citric buffer (pH 6.0) at 95 °C for 5 min. The TMA slides were blocked with 4% bovine serum albumin (BSA) in PBS for 30 min, incubated with primary antibodies at 4 °C for 12 h, and washed with PBS thrice. Anti-rabbit/mouse-horseradish peroxidase (HRP) (Dako, Carpinteria, USA) was then applied for 1 h. The slides were developed using diaminobenzidine (DAB) and Chromogen mixture (Dako, Carpinteria, USA). Nuclear counterstaining was performed out with Mayer's hematoxylin (Sigma-Aldrich, St. Louis, MO, USA). The TMA slides were dehydrated in different concentrations of ethanol (70, 80, 90, and 100%), cleared in xylene, and mounted. The protein staining intensities were evaluated on a semi-quantitative scale of 0–2+, with 0 (none or positive) in less than 5% of cells, 1+ (moderately positive) in 5–40% of cells, and 2+ (strong positive) in more than 40% of cells. Cases were deemed positive for target proteins if more than 5% of the cells showed staining of any intensity. The numerical score was validated by a second independent examination. Primary antibodies against KRAS and DX2 were purchased from Novus Biologicals and CureBio, respectively. To evaluate whether KRAS antibody specifically detects KRAS, lysates of HeLa cells in which the KRAS gene was knocked out using CRISPR/Cas9 (Abcam) were subjected to SDS-PAGE and western blotting, and the specific recognition of KRAS by the antibody was confirmed.

## Patient analysis. Tumor and matched normal tissues of patients with colorectal cancer were obtained from Yonsei Hospital. Colon cancer tissues and adjacent normal mucosa were sampled from patients with colorectal cancer who underwent surgical resection at Severance Hospital Yonsei University Health System. Written informed consent was obtained from the patients before sampling, according to the Declaration of Helsinki. The sampling protocol was approved by the Severance Hospital Yonsei University Health System Institutional Review Board (IRB approval no. 4-2016-0406). All tissues ($n = 99$) were lysed with 50 mM Tris-HCl (pH 7.4) buffer containing 100 mM NaCl, 0.5% Triton X-100, 0.1% SDS, 10% glycerol, 1 mM ethylenediaminetetraacetic acid (EDTA), and protease inhibitor (Calbiochem). The lysates were subjected to SDS-PAGE and western blotting using specific antibodies against DX2, KRAS, and actin. DX2 and KRAS levels were quantified and normalized to the actin levels. Normalized values for tumors and paired normal tissues were compared.

## Cell line analysis. Different cancer cell lines were lysed with 50 mM Tris-HCl (pH 7.4) buffer containing 100 mM NaCl, 0.5% Triton X-100, 0.1% SDS, 10% glycerol, 1 mM EDTA, and protease inhibitor (Calbiochem) and the extracted proteins were separated by SDS-PAGE and subjected to western blotting analysis using the antibodies specific to DX2 and KRAS. The band intensities of DX2 and KRAS in western blots of the tested cells were measured using FUSION FX (Vilber). Among them, the quantitated maximum and minimum values were shown with the highest and lowest color intensity, respectively, and the remaining values were graded according to their relative values.

## Surface plasmon resonance analysis. The interaction between DX2 and KRAS4B proteins was analyzed at 25°C using a surface plasmon resonance instrument (SR7500 DC, Reichert Inc., NY, USA). DX2 protein was kindly provided by Dr. Young Ho Jeon (Korea University), and KRAS4B protein was purchased from Abcam. KRAS4B was immobilized on a 500,000 Da carboxymethyl dextran hydrogel surface sensor chip (Reichert Technologies, Depew, NY) by a reaction with a mixture of KRAS4B (3 μg/mL) and immobilization buffer containing 10 mM sodium acetate (pH 5.0). BSA was used as a control for KRAS4B and immobilized using the same procedure as for KRAS4B at a rate of 20 μL/min for 10 min. Immobilization levels of KRAS4B and BSA were 4300 and 7500 RU, respectively. Different concentrations of DX2 (0.1–3.2 μM) in PBS binding buffer (pH 7.4) were flowed over the surface of chip with immobilized KRAS4B at a rate of 30 μL/min for 5 and 6 min of association and dissociation time, respectively. The sensor surface was regenerated after each association and dissociation cycle by injecting 10 mM NaOH for 1 min. Sensorgrams were fitted to a simple 1:1 Langmuir interaction model ($A + B \rightleftharpoons AB$) using the data analysis program Scrubber 2.0 (BioLogic Software, Australia, and Kaleida Graph Software, Australia)[97].

## Fluorescence-based equilibrium binding assay. All titration experiments for determining the binding affinity between DXI and DX2 proteins were conducted at 20 °C using a Jasco FP 6500 spectrofluorometer (Easton, MD, USA). Purified human tag-free DX2 proteins were equilibrated with different concentrations of DXI before the fluorescence emission was measured. Ligand stock solutions were titrated into a protein sample dissolved in phosphate buffer (pH 7.4) containing 137 mM NaCl, 2.7 mM KCl, 10 mM $Na_2HPO_4$, and 2 mM $KH_2PO_4$. Protein samples were excited at 280 nm, and the decrease in fluorescence emission upon

ligand binding was measured at 348 nm as a function of the ligand concentration. All titration data were fitted to a hyperbolic binding equation to obtain $K_d$ values.

**Quantitative co-immunoprecipitation.** Each RAS isoform was cloned at the EcoRI/XhoI sites of the pC[Nluc/MCS/CMV/Neo] (Kan) vector. CCD18CO cells expressing nanoluciferase-RAS and Strep-DX2 were lysed with 50 mM Tris-HCl (pH 7.4) buffer containing 100 mM NaCl, 0.5% Triton X-100, 0.1% SDS, 10% glycerol, 1 mM EDTA, and protease inhibitor (Calbiochem) and centrifuged. The supernatants were precipitated with a strep-tactin column (IBA), following the manufacturer's instructions. The luciferase activities of the eluted proteins were normalized to those of the whole cell lysates after excluding the background luciferase activities of the control in which Strep-DX2 was not expressed. Luciferase activity was measured using the nanoluciferase assay system, following the manufacturer's protocol (Promega). The experiments were independently repeated thrice.

**Quantitative in vitro pull-down assay.** GST-tagged proteins were incubated with nanoluciferase-RAS proteins in 293T cell extracts in 50 mM Tris-HCl (pH 7.4) binding buffer containing 100 mM NaCl, 0.5% Triton X-100, 10% glycerol, 1 mM EDTA, and protease inhibitor (Calbiochem). After incubation for 4 h at 4 °C, GST proteins were precipitated with glutathione-Sepharose beads and washed with binding buffer thrice. The amounts of the proteins co-precipitated with GST proteins were measured using the luciferase activity (Promega). The activities were then normalized to those of the extracts after removing the background activity bound to GST-EV. The experiments were independently repeated thrice.

**Gel filtration chromatography.** Strep-DX2- and -AIMP2-transfected CCD18CO cells were treated with EGF for 4 h and lysed with 50 mM 4-(2-hydroxyethyl)-1-piperazineёthanesulfonic acid (HEPES) (pH 7.6) buffer containing 300 mM NaCl, 1 mM EDTA, 10% glycerol, 0.5% Triton X-100, and 1 mM DTT for size-exclusion chromatography. The cell lysates were filtered using a 0.22 μm syringe filter. Filtered lysates (1 mg) were loaded onto the column (Superdex 200 10/300 GL, GE Healthcare) in AKTA FPLC system. Elutes at a flow rate of 0.4 mL/min from loaded lysates were subjected to SDS-PAGE and western blotting using anti-Strep antibody to detect DX2 and AIMP2.

**Ubiquitination assay.** The cells were treated with DXI in a dose-dependent manner and MG-132 (50 μM) for 18 h. The cells were lysed with 50 mM Tris-HCl (pH 7.4) buffer containing 100 mM NaCl, 0.5% Triton X-100, 0.1% SDS, 10% glycerol, 1 mM EDTA, and protease inhibitor (Calbiochem). After evaluating the specificity to detect KRAS, not NRAS and HRAS, proteins by KRAS antibody (Santa Cruz), we used anti-KRAS antibody for immunoprecipitation[98]. Endogenous KRAS was immunoprecipitated from the cell lysates with a specific antibody, and the precipitated proteins subjected to SDS-PAGE. The amounts of ubiquitinated KRAS proteins were determined by immunoblotting with an anti-ubiquitin antibody (Santa Cruz). For the in vitro ubiquitination assay, 150 ng of E1 (UBE1, BostonBiochem), 150 ng of E2 (UbcH5c, BostonBiochem), 100 ng of E3 ligase (Smurf2, LSBio), 10 μg of ubiquitin (BostonBiochem), and 200 ng of substrate (KRAS4B, Abcam) were mixed in reaction buffer (30 μL) containing 100 mM ATP-Mg. After incubation at 30°C for 60 min, half of the mixture was precipitated at 4°C for 4 h by using anti-KRAS antibodies. Precipitated KRAS4B was subjected to SDS-PAGE and immunoblotting using an anti-ubiquitin antibody to detect the ubiquitin-conjugation of KRAS4B. The other half of the mixture was subjected to SDS-PAGE and Coomassie staining to confirm protein composition.

**In vitro farnesylation assay.** KRAS4B protein (5 μM) was used as a substrate and incubated with 5 μM FTase (Jena Bioscience) and 25 μM NBD-GPP (Jena Bioscience) in prenylation buffer (50 mM HEPES pH 7.2, 50 mM NaCl, 5 mM DTT, 5 mM MgCl₂, and 20 μM GDP [Sigma]) in a final volume of 20 μL[99]. The reaction mixtures were incubated for 15 min at room temperature and then quenched by the addition of 20 μL of hot 2X SDS sample buffer. The samples were boiled at 95 °C for 3 min, and 25 μL was loaded onto a 12% SDS-PAGE gel. The fluorescent bands corresponding to the farnesylated proteins were visualized in the gel using a Fluorescent Image Reader FLA-5000 (Fuji).

**In vivo analysis.** To check whether DX2 levels affect KRAS-mediated tumorigenesis in vivo, we generated stable cells expressing various levels of DX2 and KRAS. H460 cells were transfected with DX2-specific shRNA-expressing pLKO.1 and selected with puromycin (1 μg/mL). The selected cells were transfected with KRAS4B-expressing pEGFP-C1 and subjected to a second selection with G418 (800 μg/mL, Duchefa). After sequential antibiotic selection, the expression of DX2 and KRAS4B was checked by immunoblotting using specific antibodies against DX2 and GFP, respectively. H460 stable cells ($2 \times 10^7$) were subcutaneously injected into two sites (left/right) of the backs of 7-week-old female BALB/cSLC-nu/nu mice (Central Laboratory Animal) ($n = 4$ per group). Mice were housed under ambient temperature of 24 ± 2 °C, circulating air, constant humidity of 50 ± 10% and a 12 h:12 h light/dark cycle. After three d of injection, tumor size and body weight were measured five times for the 15 d. To measure the in vivo efficacy of DXI, H460 cells ($1 \times 10^7$) were xenografted as described above. After seven d, 1 or 5 mg/kg of DXI was injected intraperitoneally, and the volume of the embedded tumors and body weight of mice were measured five times for 12 d. HCC1588 and A549 ($1 \times 10^7$) cells were also used for xenograft experiments, as described above. After three d, 5 mg/kg of DXI was intravenously injected five times during the experimental period. To check the SABV (sex as a biological variable), H460 ($1 \times 10^7$) cells were xenografted to 7-week-old male or female BALB/cSLC-nu/nu mice and DXI (5 mg/kg) were administered intravenously as described above. All mice were sacrificed, and the embedded tumors were isolated to measure their sizes and weights. The tumor volume was calculated using the formula: $V = 1/2a^2b$ (V: volume, a: shortest diameter, b: longest diameter). This study was reviewed and approved by the Institutional Animal Care and Use Committee (IACUC) of Seoul National University (SNU). We confirm that all experiments were performed in accordance with relevant guidelines and regulations. The maximum allowable tumor size by IACUC of SNU is 20 mm in diameter and we confirmed that the maximal tumor size was not exceeded.

**Anchorage-independent colony formation assay.** Cells ($5 \times 10^5$) expressing Strep-DX2 or GFP-KRAS4B were subjected to anchorage-independent colony formation assay using a cell transformation assay kit (Cell Biolabs, Inc.), following the manufacturer's instructions. After 10 d, settled colonies were stained with 3-(4, 5-cimethylthiazol-2-yl)-2, 5-diphenyl tetrazolium bromide (MTT) solution (Sigma), and the number of colonies was counted. The experiments were independently repeated thrice.

**MTT assay.** Cells ($1 \times 10^4$) were cultured in 96-well plates for 24 h, and DXI was added at different concentrations for 72 h. MTT solution (10 μL; 5 mg/mL, Sigma) was added to 100 μL of medium in each well, and cells were incubated for 30 min at 37 °C. The precipitated formazan crystals were dissolved in 100 μL DMSO (Duchefa). Absorbance was measured at 420 nm using a microplate reader (Sunrise, TECAN).

**RT-PCR.** Total cellular RNA was extracted using an RNeasy Mini Kit (Qiagen) and subjected to RT-PCR with dNTP, random hexamer, and Moloney murine leukemia virus. cDNA (1 μL) was used for PCR to determine the expression of KRAS, DX2, and actin using the following specific primers: KRAS: ACAGGCTCAGGACTTA GCAAGAA and AGGCATCATCAACACCCTGT; DX2: CTGGCCACGTGCAGG ATTACGGGG and AAGTGAATCCCAGCTGATAG; Actin: CCTTCCTGGGC ATGGAGTCCT and GGAGCAATGATCTTGATCTT.

**Statistics and reproducibility.** Statistical analyses were performed using Prism (GraphPad). Student's two-tailed t test was performed for statistical analysis. Statistical significance was set at $P < 0.05$. All error bars represent the standard deviation (S.D.). To represent the results as a heat map, the quantitated maximum and minimum values were shown with the highest and lowest color intensity, respectively, and the remaining values were graded according to their relative values. All experiments were repeated independently with similar results for three times and the representative data were shown.

**Reporting summary.** Further information on research design is available in the Nature Research Reporting Summary linked to this article.

## Data availability

All data are available within this article and the Supplementary Information files. The mass spectrometry proteomics data generated in this study have been deposited in the ProteomeXchange Consortium via the PRIDE[100] partner repository under accession code PXD029839. Source data are provided with this paper.

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

## Acknowledgements

We thank Dr. Jeong Kyu Bang of Korea Basic Science Institute (KBSI) for providing KRAS4B HVR peptide. This work was supported by the National Research Foundation of Korea (NRF) grant funded by the Korea government (MSIT) (NRF-2021R1A3B1076605), IMRCTR grant (NRF-2018R1A5A2023127) of National Research Foundation, SI-1951-30 funded by the Ministry of Science and ICT (MSIT) of Korea, and Yonsei University Research Fund of 2020-22-0356.

## Author contributions

D.G.K., Y.C., K.L[1]., and S.K. conceptualized the study; D.G.K., Y.C., Y.H.J., S.I[2]., and K.L[1]. designed the methodology; D.G.K., Y.L., S.I[3]., J.K., J.S., Y.R., J.G., H.Y.C., A.U.M., J.L., S.H.P., and D.K. performed the experiments; D.S.H., and K.I[4]. synthesized the chemicals; B.S.M., and K.Y.L. provided the clinical specimens; D.G.K., Y.C., Y.H.J., S.I[2]., K.L[1]., and S.K. performed the analysis; D.G.K., K.L[1]., and S.K. wrote the paper; S.K. provided overall supervision. All authors reviewed and approved the final paper. (K.L.[1]: Kyeong Lee, S.L.[2]: Sunkyung Lee, S.L.:[3] Semi Lim, and K.L.:[4] Kwanshik Lee).

## Competing interests

The authors declare no competing interests.
