## [Peer Review File · Nature Communications]

AIMP2-DX2 provides therapeutic interface to control
KRAS-driven tumorigenesisEditorial Note: Parts of this Peer Review File have been redacted as indicated to maintain the confidentiality of unpublished data.

REVIEWER COMMENTS

Reviewer #1 (Remarks to the Author): with expertise in Ras biology and computational drug discovery

Journal: Nature communications

Manuscript ID: NCOMMS-21-19178-T

Type of manuscript: Article

Title: AIMP2-DX2 provides unique therapeutic interface to control KRAS-driven tumorigenesis

Authors: Dae Gyu Kim, Yongseok Choi, Yuno Lee, Semi Lim, Jiwon Kong, JaeHa Song, Younah Roh, Dipesh S. Harmalkar, Kwanshik Lee, Ja-il Goo, Hye Young Cho, Ameerq Ul Mushtaq, Jihye Lee, Song Hwa Park, Doyeun Kim, Byung Soh Min⁶, Kang Young Lee, Young Ho Jeon, Sunkyung Lee, Kyeong Lee, and Sunghoon Kim

Corresponding Author: Kyeong Lee and Sunghoon Kim

Received: May 2021

Summary: In this paper by Kim et al., the relationship between the protein AIMP2 (Aminoacyl tRNA synthase complex-interacting multifunctional protein 2, a tumor suppressor) and the RAS proteins is examined; specifically, DX2, which is a splicing variant of AIMP2 that lacks exon II and is upregulated in many cancers, is shown to bind with KRAS4b. The Authors describe (1) that DX2 enhances protein stability of KRAS; (2) that DX2 binds with specificity to KRAS4B (at a Kd of 151 nM by SPR), (3) the characterization of DX2-KRAS complex (4) that DX2 inhibits Smurf2-mediated ubiquitination of KRAS; (5) the identification and characterization of a DX2-KRAS inhibitor, BC-DX1-32982, which binds to DX2 with IC50 of 240 nM; and (6) Investigation of mode of action of BC-DX1-32982. The authors use several experimental methods to interrogate the relationship between DX2 and KRAS including on purified protein, in cells, and in mouse models with xenografts, as well as performing a chemical screen to identify a lead molecule that disrupts the interactions between DX2 and KRAS. In addition, the authors use calculations to complement the experiments and make predictions that they tested experimentally. I would recommend publication. The authors may considered the following suggestions/comments:

(1) I was not able to open the Supplementary Movie 2 (312385_0_video_352157_qsr7qr), I suggest that this be addressed.

(2) KRAS is the second most populated from the "potential binding proteins of DX2 by LC-MS analysis", but just for the Proliferation ontology, KRAS is ranked 89 among all proteins identified. It might be worth digging in and discussing these other proteins and this figure more as it has rich information. What was the first Proliferation ontology? (PRDX1?) and what was third of Proliferation ontology? (VCP?). What was the difference in population?

(3) Please describe the SPR experiment in more detail (so that it can be reproduced).

(4) I am interested in how the binding of AIMP2 full length protein to KRAS compare to that of the DX2 slicing variant to KRAS. Does the full-length protein not bind?

(5) Could the authors clarify what they mean by restraint force do they mean the they are applying a restraint to the proteins? If so, what residues are they restraining, are they just restraining the backbone (what is the restraint mask)? Why is this restraint needed for all 500ns if the simulation is stable for the last 100 ns? What happens if you take a snapshot from the first 100ns and restart the simulation from that point without restraints? Oh. I see that the authors explain more about using upper-wall restraints as implemented in the Plumed plugin version 2.448 with GROMACS software (version 2016). Could the authors direct the reader to this information in the result section by adding something like "see methods subsection Molecular docking simulation. Preparation of the systems. for more details" I worry that the restraints is biasing your results and wonder how your results might differ without the restraints. Could the authors add more detail on why these restraints are needed and how, during their interpretation of the results, they were convinced that this bias is helpful and not deceiving (e.g. NMR result comparison?).

(6) Could the authors add some plots to the Supplemental figures showing simulation behavior. The movies are helpful but are qualitative.

a. the RMSD of the two proteins over time for the DX2, RAS simulation and the RMSD of ligand for the DX2, BC-DX1-32982 simulation.

b. Could they produce a figure containing plots that show key distances over time for example the

salt bridges and hydrogen bonds shown in figure 3. Could they also show the interacting pairs so that it is clear which residues from KRAS are interacting with which residues of DX2.

c. The authors state “We found that KRAS4B HVR anchored to the site near H86 of DX2 within 10 ns, and maintained the averaged distance of ~11 Å of C-alpha atoms between DX2 H86 and KRAS4B I187 during the entire simulation time.” Plotting this distance would be very helpful; this distance might be helpful in understanding the behavior of the simulations.

d. Measuring the RMSD of the HVR region might also be helpful.

(7) On page 8, line 164 and 165, the authors state: “We then generated alanine-substitution mutants at the residues predicted to be crucial for binding and examined their effects on the interactions by immunoprecipitation.” How were these alanine mutant residues picked? Were they chosen through the molecular dynamics simulation? what were the criteria for the authors to select a residue as important? Were interaction energies (using molecular mechanics, free energy calculations) used?

(8) Could molecular modeling/simulation help understand how the alanine mutant is impacting binding of DX2 and KRAS4B?

(9) On page 8, lines 171 and 172, The authors state “These results are consistent with those of NMR-based CSP analysis,” could the authors quantify this? or state how they are consistent?

(10) On page 9 line 187, the authors reference Extended Data Fig 3d, is this the right figure number? Please explain more.

(11) On page 12, lines 252-254, this sentence is confusing, should “three” be “free”.

(12) The IC50 determination of BC-DXI-32982 binding to DX2 is 240 nM. Could the authors confirm this binding affinity for the small molecule using an alternative assay like SPR to determine the affinity? or another method. I would prefer to see a Kd of binding for the small molecule to DX2. Or perhaps they could determine the Ki using the nano BRET assay? What is the binding affinity of the small molecule to the AIMP2 (not the splice variant)? Could the affinity to the full protein AIMP2 explain the activity seen to cells without DX2 (Fig 6e and Extended Data Fig. 12 h). What is the impact of the small molecule on the AIMP2 binding partners?

(13) Could the authors confirm the impact on binding of the alanine mutations with simulations. For example, looking at pose stability.

a. Could the authors show RMSD plots showing the pose stability? Perhaps showing the stability of the simulation using the proposed model displayed in Fig 6. Could they run multiple (three) replica simulations (no production restraints) starting from the proposed model in Fig 6 and show that this pose is stable.

b. Perhaps the authors could introduce the alanine mutations (e.g. Y47A) and confirm that it destabilizes the complex.

(14) The authors state: “Although recent advances in targeting KRAS show some promising results.” This should be updated with the recent news the Amgen G12C inhibitor is now an approved drug. They may also like to adjust the abstract.

(15) On page 30 , line 718, In the method section, the authors refer to “unpublished data”? why not publish the details of the simulation in this publication? Or are the authors planning on publishing in a future manuscript?

(16) Why use a H-RAS structure bound to SOS for your modeling?

(17) Text modifications:

a. On page 3, line 57, add the word cells, that is, change “H460 lung cancer” to “H460 lung cancer cells”

(18) It would be helpful if abbreviations are defined in the figure legend (this is needed mostly in the supplemental figure captions), Like: EV, IP, WCL, sh-DX2, sh-con, etc. particularly for a general reader without familiarity with all of these methods.

(19) Do KRAS inhibitors, like BI-2852 impact binding of DX2 to RAS? Have the authors considered this?

Reviewer #2 (Remarks to the Author): with expertise in Ras biology and cancer

In this manuscript, Kim et al. identified a specific interaction of KRAS and DX2, which prevented ubiquitin-mediated degradation of KRAS protein prior to farnesylation. By genetic and pharmacological targeting this interaction, they show that the proliferation in vitro and tumor

growth in vivo is impaired. While by far most of the RAS researchers have focused on the regulation of KRAS activity through its G domain, the significance of their work is to define the oncogenic function of DX2 in regulation of KRAS4B protein level through the binding to HVR domain. Interestingly, such regulation works across several different KRAS mutants, including G12C, G12D, and Q61R, which may have a broader clinical impact to treat cancer patients harboring KRAS mutations. However, there are some major concerns listed below.

1. Based on the data in Fig 1 and 2, DX2 binds to both KRAS4A and 4B, but not N- or H-RAS. It is surprising since the structure of KRAS4A HVR region is notably distinguished from KRAS4B and is more similar to N- and H-RAS in many prospects, especially farnesyl-modification. The authors however failed to address this in the following figures 3 & 4 (they only use KRAS4B protein to validate the binding to DX2), which also dimmed the biological importance, since it has been increasingly appreciated that KRAS4A and 4B have different biochemical functions in cancers.
2. Based on their hypothesis and data listed in Figure 1-2, DX2 expression level and binding presumably only affect the protein level of KRAS in a GTP independent manner. It is somehow contradictory to their interactome data (figure 2A). In the setting of the experiment, they identified DX2 interacting protein in EGF treated H460 cells. H460 cell line is known to be "KRAS mutant", and EGF stimulation should only affect the RAS-GTP loading but not protein level. The authors then failed to address the effects of GTP binding in DX2-KRAS interaction in their protein-protein docking models. The lack of such data also weakens the importance of their finding, since KRAS-WT (which has higher binding affinity to GDP without stimulation) is known to have suppressive effect to the oncogenic activity of mutant KRAS. In addition, the protein level of KRAS (which they majorly focused on here) cannot completely translate to its oncogenic functions and activity.
3. H460 is reported to be somehow KRAS independent (defined by Jeff Settleman's group; Cancer Cell 15, 489–500; June 2, 2009). It is therefore somehow surprising that their compound had such impact in H460 xenografts in Fig 5H. More cell lines with various KRAS-GTP level and/or dependency will strengthen the data.
4. IHC in Fig 5H is hard to define the co-localization of KRAS and DX2. Also, it is known that there are no antibodies ideal for KRAS IHC. Yet, the information of the KRAS antibodies that they used here cannot be found in the manuscript. Positive and negative controls are therefore needed to validate their IHC.
5. A lot of important experiment details are missing. For example, the information of transgenic mice where DX2 expression can be induced by dox (line 60) is missing. No reference was provided either. On the similar note, the information of nanoluciferase-RAS constructs used in Figure 2b was not provided. Phosphorylation site of Akt in all western blots was not labeled.

Reviewer #3 (Remarks to the Author): with expertise in aminoacyl-tRNA synthetases

Nature Communications manuscript NCOMMS-21-19178-T

AIMP2-DX2 provides unique therapeutic interface to control KRAS-driven tumorigenesis

Kim, Choi, et al identified an interaction between DX2, a splice variant of the multifunctional multisynthetase complex adapter protein AIMP-2, with the master oncogene KRAS. The interaction protected KRAS against ubiquitination by the E3 ligase Smurf2 and could be mapped to a specific protein-protein interface. The authors developed a small molecule inhibitor that disrupts the KRAS-DX2 interface and treatment with this inhibitor successfully inhibited tumor growth.

The authors did a thorough job on verifying their initial observations and characterizing the interaction on a molecular level. Their experiments confirm their findings, and the main observation is exciting and relevant. However, in its current state, the results are difficult to understand, and the presentation must be improved.

Major issues concerning data presentation and manuscript structure:

- 1) The introduction is barely existent and does not provide adequate background information for the many aspects investigated in the result section. There are 2 sentences on KRAS, 2 sentences on DX2.

2) In the discussion, if statements refer to findings within the manuscript, the corresponding figure should be cited at the end of the sentence. If sentences refer to a previously described finding, the corresponding literature should be cited at the end of the sentence. Overall, the discussion could expand more on the context of the findings rather than repeating the result section and would benefit from language editing.

3) A lot could be done to make it easier for readers to understand the figures. For example, Fig 4g – it is not clear that this is IP until one reads the figure legend and it is not clear that the time refers to time after EGF addition until one looks at Ext. Figure 8a and b. What is meant by relative binding here? The same applies to several of the main figures where just looking at the figure, it is not clear what the main message, experimental system, or focus of the figure should be. Of course, figure legends are always necessary, but it would help a lot with comprehension if the figures would be more self-explanatory. Clearly, the authors put lots of thought and work into their experiments - making the data more accessible would help enormously for the community to appreciate their findings.

4) Protein sizes are annotated inconsistently in western blots and gels, as are fold-changes. It should be noted how often experiments were repeated throughout and if technical or biological replicates were used. n numbers for animal experiments must be noted in the figure legends. There are no statistics on anything. It is appreciated how clean some of the results are, but still, statistics and quantification are necessary.

Major issues with experimental data:

1) The interactome of DX2 shown in Figure 2a contains many proteins. The method section does not mention if a fold-change over EV was used to determine true interactors over contaminants. The relative abundance in the corresponding table is < 1 for all interactors (DX2/EV) – if that is correct there is no enrichment over EV? Additionally, while KRAS is unarguably an important protein, it is far from the most enriched. Are the other interactions competing with KRAS for DX2 binding?

2) The predicted interaction interface of DX2/KRAS4B is mostly located in exon 3 – how would the insertion of exon 2 in AIMP2 disrupt the interaction to achieve the specificity that is shown for DX2 over AIMP2?

3) Is total ubiquitination affected by compound 32982? At least it should be shown that ubiquitination of other proteins are not affected by 32982 to confirm that 32982 is not globally upregulating ubiquitination.

4) What is the proportion of endogenous DX2 to full-length AIMP2 in the cell lines that were used? Is there expected to be enough DX2 to stabilize KRAS?

Minor issues:

1) Are any KRAS mutations found in cancer that are predicted to localize to the interact of the KRAS/DX2 interaction that would stabilize/destabilize the interaction?

2) Figure 2e, the DX2 blot looks odd as the background is a lot lighter on one side.

3) Fig 4g – why does the red line peak at 20 minutes when there is no time point measured and 10 and 30 minutes have the same relative binding? Same for 40 minute time point on the right panel, also red line.

4) Fig 5h, IHC should contain a scale bar.

5) Fig 11a – does 32982 display toxicity on non-KRAS dependent, untransformed cells, for example MEFs?

6) How was the DX2 specific siRNA generated – does it span the exon 1 – exon 3 junction?

7) Ext Fig 2d: Why has the KRAS antibody been verified but not the DX2 antibody? The whole membrane should be shown here as unspecific bands with molecular weight other than the predicted molecular weight will give signal in IHC.

7) Ext. Figure 12e has a typo in FLAG. Figure 4c has a typo in Smurf2. Ext Fig 7d and e also have typos in Smurf2.

Reviewer #4 (Remarks to the Author): with expertise in structural biology, tRNAs

Review of the manuscript Nature Comm by Dae Gyu Kim et al.

In this work, the authors provide a comprehensive study of the role of DX2, a splicing variant of the scaffold protein AIMP2 involved in mu, on KRAS-driven tumorigenesis. In several cancers such as colorectal and lung cancer, AIMP2 and DX2 seems to positively correlate, thus offering opportunities for therapeutic intervention. However, most efforts to block KRAS driven tumorigenesis have not been very rewarding so far. By an elegant study, the authors find that DX2 but not AIMP2, associate directly to oncogenic KRAS proteins, enhancing their stability and augmenting tumorigenesis. Moreover, the authors identify the exact interacting region of DX2 to KRAS, and the spatiotemporal basis of this interaction in the cell in the cell (cytosol). The binding of DX2 to KRAS appear to block ubiquitin-mediated degradation of KRAS by Smurf2. Besides this very important advance in our knowledge on the role of DX2 in controlling KRAS oncogenic functions, the authors go beyond it and identify a potent inhibitor (~0.2 μ M) of the interaction DX2-KRAS, which combined with MD studies and mutational analysis suggests that it blocks the binding interface of DX2 to KRAS, thereby likely exposing KRAS to Smurf-2 mediated degradation. Not only the authors show that this compound can arrest growth of cancer-derived cell lines, but also show that it has efficacy to decrease size tumours in mice xenografts studies. Collectively, this study proves that inhibition of KRAS-DX2 interaction is a and novel and very promising approach to develop anticancer therapeutics in patients with high levels of KRAS.

The manuscript is written in a very clear way and in general the interpretation of the results and discussion is very well justified.

After correcting some small issues, I would recommend this work for publication:

- Both AIMP2 and its splicing variant DX2 contain the GST domain, which is important for the assembly of several components of the multi-synthetase complex (MSC) via AIMP2. Since this GST domain in DX2 appears to be important for the binding of the inhibitor 32982, it is very intriguing that it does not do the same with AIMP2. Although the authors briefly discussed this point in lines 359-360, it is not clear how this could be possible. Also the cellular levels of components of the MSC are not affected by the inhibitor 32982, but this does not prove that the compound does not bind to AIMP2. The binding of 32982 to AIMP2 might occur without impacting the cell levels of proteins of the MSC. Another interesting question is : Does the putative binding of 32982 prevents the assembly of MSC mediated by AIMP2 ?.

- Figure 1 F lacks the units of the scale bar 0-10. What arbitrary units were chosen to assign the values 0 to 10?

- As a structural biologist, a general comment about the manuscript is that I would have loved to see real structures of DX2 bound to KRAS, and how the binding of the inhibitor 32982 would block such interaction. Of course, the authors might agree with me that this is beyond this study, but it would certainly be very exciting to see this in a future study. Indeed, the identification of the exact binding regions of KRAS to DX2, and the relative high affinities as measured by SPR, would suggest that crystallography studies are feasible. Nevertheless, I agree with the authors that the disordered nature of KRAS HPV would create hurdles for crystallization, thus I suggest to use short peptides of this region of KRAS to try co-crystallization with DX2.

- Lines 147-150: the authors provide an interpretation about the significant line broadening and disappearance of peaks in the NMR spectra due to induced non-specific aggregation of DX2.

Besides this explanation, it is also possible that there is binding and that this is in an intermediate exchange, likely in the μ s to ms timescale, which could not be detected in these NMR experiments.

- Lines 228-230: "Smurf2 is the ligase..." The authors did not investigate the other ligase (B-

TRCP), therefore "the ligase " should be modified to "a ligase" ?. also in line 230, should not be written " ...SMURF2 could function as the ligase ...".

Other minor corrections:

- Please revise grammar of phrase in lines 103-106.
- Line 185: CRE1- mediated cleavage or RCE1 ? Please check this is correct.
- Phrase in lines 212-215 appears incomplete ?
- Typo in panel 4 c: there should be Smurf2 and not Smruf2.
- Typo in line 247: "...conditions were determined...".
- Line 250: " ...two different binding modes...."
- Please revise gramma of phrase at line 252-257.
- Line 293: "...BC-DXI-32982 is a specific...."
- Figure 5g: bar scale 0-10 lacks units, μM ?
- Panel 5h (right) is not cited in the text ?
- please revise grammar of lines 329-330.
- Line 378: "...but these two residues..." ?
- Line 381: remove "...and...".
- Line 383: correct "did not affected..."
- Not sure it makes sense phrase at line 389-392.
- Please revise phrase at lines 395-397.

Reviewer: Dr Andrés Palencia,
National Institute of Health and Medical Research,
Institute for Advanced Biosciences,
Grenoble, France
website: <https://noveltargets-palencia.com>

Reviewer #5 (Remarks to the Author): with expertise in chemical probes

This manuscript reports that AIMP2-DX2, a variant of tumor suppressor, AIMP2, acts as a cancer-specific regulator of KRAS stability, augmenting KRAS-driven tumorigenesis. A small molecule that specifically binds to the KRAS-binding region of AIMP2-DX2 and inhibits the interaction of the two factors was also identified. These results suggest the interface of AIMP2-DX2 and KRAS as a unique route to control KRAS-driven cancers. Overall, the findings are interesting and novel. This reviewer recommends the acceptance for publication after the following concerns addressed.

- 1) Since human tissues are used, the compliance with the ethics and government regulations should be clearly stated in the manuscript.
- 2) The in vivo animal studies were limited on only female mice. The SABV (sex as a biological variable) should be considered.
- 3) The in vivo animal studies of the inhibitor only used $n = 4$ per group, which is a little bit low. Also, more cell lines using KRAS wild type and mutant type should be conducted for the in vivo efficacy comparison for the inhibitor.
- 4) The in vivo toxicity studies of the inhibitor should include the blood analysis and histology changes of various organs.
- 5) It is not clear how the dosages for testing the in vivo efficacy were chosen. The in vivo PK studies including the half-life and drug exposure, etc. should be analyzed.
- 6) The binding affinity and IC50s should have \pm SD values for the variation.
- 7) The off-target effects of BC-DXI-32982 are inadequately characterized. The target specificity of the reported inhibitor using a wide panel of drug targets (e.g., Eurofin drug target panel assays) should be pursued.
- 8) For the biotin tool compounds, the key moiety of quinoxaline in the parental inhibitor has been removed in the design of biotin-labeled tool molecules. Not sure how close these biotin tool compounds actually reflect the parental inhibitor in terms of the target engagement.

Answer letter (Re: NCOMMS-21-19178-T)

AIMP2-DX2 provides unique therapeutic interface to control KRAS-driven tumorigenesis

Dae Gyu Kim^{1,7}, Yongseok Choi^{2,7}, Yuno Lee³, Semi Lim¹, Jiwon Kong¹, JaeHa Song¹, Younah Roh¹, Dipesh S. Harmalkar^{2,4}, Kwanshik Lee², Ja-il Goo², Hye Young Cho⁵, Ameerq Ul Mushtaq⁵, Jihye Lee¹, Song Hwa Park¹, Doyeun Kim¹, Byung Soh Min⁶, Kang Young Lee⁶, Young Ho Jeon⁵, Sunkyung Lee³, Kyeong Lee^{4,*}, & Sunghoon Kim^{1,8,*}

¹Medicinal Bioconvergence Research Center, Institute for Artificial Intelligence and Biomedical Research, College of Pharmacy & College of Medicine, Gangnam Severance Hospital, Yonsei University, Incheon 21983, Korea. ²Department of Biotechnology, Korea University, Seoul, 02841, Korea. ³Drug Information research Center, Korea Research Institute of Chemical Technology, Daejeon 34114, Korea. ⁴College of Pharmacy, Dongguk University, Goyang 10326, Korea. ⁵College of Pharmacy, Korea University, 2511 Sejong-ro, Sejong 30019, Republic of Korea. ⁶Department of Surgery, College of Medicine, Yonsei University. ⁷These authors contributed equally: Dae Gyu Kim, Yongseok Choi.
*Correspondence: (K.L.), (S.K.)

Referee #1:

Q1-1) I was not able to open the Supplementary Movie 2 (312385_0_video_352157_qsr7qr), I suggest that this be addressed.

Answer: We newly uploaded the file for Supplementary Movie 2. We hope that this file is accessible to the reviewer.

Q1-2) KRAS is the second most populated from the “potential binding proteins of DX2 by LC-MS analysis”, but just for the Proliferation ontology, KRAS is ranked 89 among all proteins identified. It might be worth digging in and discussing these other proteins and this figure more as it has rich information. What was the first Proliferation ontology? (PRDX1?) and what was third of Proliferation ontology? (VCP?). What was the difference in population?

Answer: We agree with the reviewer in that the database might contain rich information on the potential diverse regulatory or pathological roles of DX2 via its interactors. Among the 12 potential binding proteins of DX2 in proliferative ontology, we focused KRAS for further functional investigation because we tested the DX2 interactome analysis under EGF signal, which is known to control the KRAS activity. PRDX1(Peroxiredoxin-1), the first ranker among the potential DX2 interactors, plays a role in cell protection against oxidative stress by detoxifying peroxides¹, and VCP (Transitional endoplasmic reticulum ATPase), the third ranker, is necessary for the fragmentation of Golgi stacks and reassembly during mitosis and also involved in DNA damage response². We checked whether PRDX1 would also interact with DX2 and be affected by EGF signal, and found that it could bind to DX2 but did not show the dependency on EGF signal. We added the result to Supplementary Fig. 3d with description in the Results (page 7, line 158) and Discussion sections (page 23, line 530).

Q1-3) Please describe the SPR experiment in more detail (so that it can be reproduced).

Answer: We added more detailed information of SPR experiment in Methods section (page 34, line 791).

Q1-4) I am interested in how the binding of AIMP2 full length protein to KRAS compare to that of the DX2 splicing variant to KRAS. Does the full-length protein not bind?

Answer: Since AIMP2 also has the KRAS-binding GST domain, it can also bind to AIMP2. We confirmed this possibility by *in vitro* pull-down assay (Supplementary Fig. 4d). However, the actual binding of KRAS to AIMP2 was not observed in cell (Fig. 2e) since AIMP2 is mainly sequestered in multi-tRNA synthetase complex (MSC) whereas DX2 exists as a free from³, being more accessible to KRAS. To further validate this possibility, we separated the proteins extracted from the cells by gel filtration and detected AIMP2 and DX2 mainly in the MSC-bound and -unbound free fractions, respectively (Supplementary Fig. 4e). Thus, KRAS would have a low chance to meet AIMP2 in cells despite the fact that it also contains the GST domain as DX2 (page 8, line 180).

Q1-5) Could the authors clarify what they mean by restraint force do they mean the they are applying a restraint to the proteins? If so, what residues are they restraining, are they just restraining the backbone (what is the restraint mask)? Why is this restraint needed for all 500ns if the simulation is stable for the last 100 ns? What happens if you take a snapshot from the first 100ns and restart the simulation from that point without restraints? Oh. I see that the authors explain more about using upper-wall restraints as implemented in the Plumed plugin version 2.448 with GROMACS software (version 2016). Could the authors direct the reader to this information in the result section by adding something like “see methods

subsection Molecular docking simulation. Preparation of the systems. for more details” I worry that the restraints are biasing your results and wonder how your results might differ without the restraints. Could the authors add more detail on why these restraints are needed and how, during their interpretation of the results, they were convinced that this bias is helpful and not deceiving (e.g. NMR result comparison?).

Answer: We described the trajectory selection process in the Results section (page 10, line 215). KRAS4B HVR anchored to the site near H86 of DX2 within 10 ns, and the restraints might not significantly change the simulation results. Since we applied the force only when the HVR loop (C α atom of T183) is more than 30Å away from DX2 (C α atom of E141) as an upper-wall restraint force, the remaining 490 ns except for the first 10 ns are the same as in normal simulation. Although the HVR loop did not come out in our result, the restraint force was applied up to 500 ns because we don't know whether this loop would escape or not when we perform. If simulation is performed without such restraint force, it will take a long time for the HVR loop to find its binding site. Those results were consistent with the results of NMR analysis, and further validated by mutational analysis.

Q1-6) Could the authors add some plots to the Supplemental figures showing simulation behavior. The movies are helpful but are qualitative.

a) the RMSD of the two proteins over time for the DX2, RAS simulation and the RMSD of ligand for the DX2, BC-DX1-32982 simulation.

Answer: Following the reviewer's suggestion, we added the plots for simulation of DX2, KRAS4B and DX2-KRAS4B to Supplementary Fig. 5e and the simulation of DX2-BC-DX1-32982 to Supplementary Fig. 13l.

b) Could they produce a figure containing plots that show key distances over time for

example the salt bridges and hydrogen bonds shown in figure 3. Could they also show the interacting pairs so that it is clear which residues from KRAS are interacting with which residues of DX2.

Answer: We made Supplementary Table 3 for the interaction pairs from the representative snapshot. We also described the key distances over time in the Results section (line 230, page 10). Please see Supplementary Fig. 5i and j for hydrogen bond and salt bridges interaction, respectively, as function of time.

c) The authors state “We found that KRAS4B HVR anchored to the site near H86 of DX2 within 10 ns, and maintained the averaged distance of ~ 11 Å of C-alpha atoms between DX2 H86 and KRAS4B I187 during the entire simulation time.” Plotting this distance would be very helpful; this distance might be helpful in understanding the behavior of the simulations.

Answer: Following the reviewer’s comments, we showed the plot in Supplementary Fig. 5f.

d) Measuring the RMSD of the HVR region might also be helpful.

Answer: Following the reviewer’s comments, we revised Supplementary Fig. 5g.

Q1-7) On page 8, line 164 and 165, the authors state: “We then generated alanine-substitution mutants at the residues predicted to be crucial for binding and examined their effects on the interactions by immunoprecipitation.” How were these alanine mutant residues picked? Were they chosen through the molecular dynamics simulation? what were the criteria for the authors to select a residue as important? Were interaction energies (using molecular mechanics, free energy calculations) used?

Answer: We have used contact surface area obtained from representative snapshot to select important binding residues (page 11, line 239) and added the supporting data for DX2 and KRAS4B to Supplementary Fig. 5l and 5m, respectively.

Q1-8) Could molecular modeling/simulation help understand how the alanine mutant is impacting binding of DX2 and KRAS4B?

Answer: We explained the selection criteria on important binding residues for the alanine mutants in Q1-7, which used contact surface area obtained from representative snapshot, and we observed that alanine mutants of DX2 or KRAS4B at the selected residues reduced the interaction each other. Thus, we think that our effort of molecular modeling/simulation was helpful to understand the binding pose between DX2 and KRAS4B.

Q1-9) On page 8, lines 171 and 172, The authors state “These results are consistent with those of NMR-based CSP analysis,” could the authors quantify this? or state how they are consistent?

Answer: The NMR and MD analysis suggested the residues at [V83, H84, T85, K90, V92, L97, W120] and [V54, L80, T82, V83, H84, T85, H86, S87, S88, K90, E94, L97, E102, P108, Q110, Y112, T117, I119, K121, K129, I132, Q133] of DX2 that would be crucial for the interaction with KRAS. Among them, [V83, H84, T85, K90] were commonly identified by both methods and the significance of [T85, K90] were further validated by mutational studies.

Q1-10) On page 9 line 187, the authors reference Supplementary Fig 3d, is this the right figure number? Please explain more.

Answer: We confirmed the cytosolic interaction of DX2 and KRAS by co-immunoprecipitation in cytosol (Fig. 3e) and cellular co-localization experiments

(Supplementary Fig. 3e). We thus referred both data in the revision to make the meaning more clear (page 12, line 262).

Q1-11) On page 12, lines 252-254, this sentence is confusing, should “three” be “free”.

Answer: To avoid confusion, we changed the sentence from “Since three independent association of DX2 including the binding with KRAS is interestingly under cellular proliferative condition” to “p14ARF and HSP70 have been previously reported as DX2-binding proteins. Thus, we checked how DX2 would encounter these proteins upon EGF signaling over time.” (page 15, line 328)

Q1-12) The IC₅₀ determination of BC-DXI-32982 binding to DX2 is 240 nM. Could the authors confirm this binding affinity for the small molecule using an alternative assay like SPR to determine the affinity? or another method. I would prefer to see a K_d of binding for the small molecule to DX2. Or perhaps they could determine the K_i using the nano BRET assay? What is the binding affinity of the small molecule to the AIMP2 (not the splice variant)? Could the affinity to the full protein AIMP2 explain the activity seen to cells without DX2 (Fig 6e and Supplementary Fig. 12 h). What is the impact of the small molecule on the AIMP2 binding partners?

Answer: We measured the binding affinity of BC-DXI-32982 to DX2 protein *via* fluorescence-based equilibrium binding assay and the K_d value was estimated as 480 nM (Supplementary Fig. 13d). We also checked direct and cellular binding of the compound to AIMP2 by *in vitro* pull-down and immunoprecipitation assay, respectively. While BC-DXI-32982 bound to AIMP2 as well as DX2 with similar degree in *in vitro* pull-down assay (Supplementary Fig. 13f), it showed positive binding to DX2 but not to AIMP2 in cellular

immunoprecipitation (Supplementary Fig. 13g). As mentioned above, AIMP2 is mainly sequestered in multi-tRNA synthetase complex (MSC) whereas DX2 exists as a free form³. Thus, DX2 in free form would be more accessible to BC-DXI-32982 than AIMP2 buried in MSC (page 19, line 452). We also examined whether BC-DXI-32982 would affect the interactions of AIMP2 with its binding partners, for instance, KARS1 (lysyl-tRNA synthetase 1) within MSC⁴ and p53 in nucleus⁵, and observed no effect (Supplementary Fig. 13h,i). We also monitored the effect of BC-DXI-32982 on the viability of untransformed MEF cells with no DX2 expression³, and found little negative activity to cell viability (Supplementary Fig. 12b). Collectively, BC-DXI-32982 appears to be effective specifically to the DX2-KRAS pair and DX2-expressing cells, with little influence on the physiological functions of AIMP2 and the viability of cells devoid of DX2.

Q1-13) Could the authors confirm the impact on binding of the alanine mutations with simulations. For example, looking at pose stability.

a) Could the authors show RMSD plots showing the pose stability? Perhaps showing the stability of the simulation using the proposed model displayed in Fig 6. Could they run multiple (three) replica simulations (no production restraints) starting from the proposed model in Fig 6 and show that this pose is stable.

Answer: We added plots (Supplementary Fig. 13j,k) and the related sentences describing the binding pose stability of the final proposed model in the Results section (page 21, line 472).

b) Perhaps the authors could introduce the alanine mutations (e.g. Y47A) and confirm that it destabilizes the complex.

Answer: We explained the destabilization of the DX2-DXI complex resulting from alanine mutation (page 21, line 487). Briefly, we introduced alanine mutation to Y47 and simulated

for ~380 ns under the same condition as DX2 WT simulation. We found the destabilization of the complex based on the results of RMSD (Supplementary Fig. 13l) and interaction energy (Supplementary Fig.13m).

Q1-14) The authors state: “Although recent advances in targeting KRAS show some promising results.” This should be updated with the recent news the Amgen G12C inhibitor is now an approved drug. They may also like to adjust the abstract.

Answer: Following the reviewer’s comments, we changed the sentence in abstract from “Although recent efforts to control KRAS-driven tumorigenesis give some promising results, direct targeting of KRAS mutants still remains challenging” to “Although direct targeting of a specific oncogenic mutant of KRAS was recently proven to be effective, controlling KRAS-driven cancers, regardless of the mutation type, remains challenging” (page 2, line 24).

Q1-15) On page 30, line 718, In the method section, the authors refer to “unpublished data”? why not publish the details of the simulation in this publication? Or are the authors planning on publishing in a future manuscript?

Answer: Simulation details were described in other work that was under review. Since it was recently published in J Pharmacol Exp Ther. 2021 (Title: Allosteric Inhibition of the Tumor-Promoting Interaction between AIMP2-DX2 and HSP70), we removed “unpublished data” and added the reference.

Q1-16) Why use a H-RAS structure bound to SOS for your modeling?

Answer: We used “1NVU, GTP-bound conformation”. The RAS isoforms do not show significant structural difference in the G-domain without the HVR. The crystal structure of

1NVU, Chain Q which has GTP-bound conformation, was mutated to KRAS4B sequence and used for molecular docking simulation (page 26, line 591).

Q1-17) Text modifications: On page 3, line 57, add the word cells, that is, change “H460 lung cancer” to “H460 lung cancer cells”

Answer: We changed the sentence from “H460 lung cancer” to “H460 lung cancer cells” (page 5, line 95).

Q1-18) It would be helpful if abbreviations are defined in the figure legend (this is needed mostly in the supplemental figure captions), Like: EV, IP, WCL, sh-DX2, sh-con, etc. particularly for a general reader without familiarity with all of these methods.

Answer: All of the abbreviations were defined in their first appearance.

Q1-19) Do KRAS inhibitors, like BI-2852 impact binding of DX2 to RAS? Have the authors considered this?

Answer: We checked whether BI-2852, the reported catalytic inhibitor of KRAS, would affect the interaction between DX2 and KRAS using NanoBiT luciferase assay. In contrast to BC-DXI-32982, BI-2852 did not inhibit the interaction of the two proteins (Supplementary Fig. 11d), further supporting the specificity of the BC-DXI-32982 activity.

Referee #2:

Q2-1) Based on the data in Fig 1 and 2, DX2 binds to both KRAS4A and 4B, but not N- or H-RAS. It is surprising since the structure of KRAS4A HVR region is notably distinguished from KRAS4B and is more similar to N- and H-RAS in many prospects, especially farnesyl-

modification. The authors however failed to address this in the following figures 3 & 4 (they only use KRAS4B protein to validate the binding to DX2), which also dimmed the biological importance, since it has been increasingly appreciated that KRAS4A and 4B have different biochemical functions in cancers.

Answer: We acknowledge the reviewer's point. However, it is the fact that DX2 binds to both of KRAS4A and 4B, but not to H- and NRAS isoforms. Although KRAS4A and 4B are distinguished especially in farnesyl modification for membrane anchoring, perhaps abundance of lysine residue in HVR of KRAS4A and 4B could commonly serve for binding of DX2. We also showed the significance of DX2 for the stability of KRAS4A and KRAS4B, not of NRAS and HRAS in Fig.1 and 2. In the following assays, we used KRAS4B as the representative KRAS isoform in Fig. 3 and 4. Since the reviewer mentioned the sequence difference between KRAS4A and 4B, we repeated some key experiments in Fig. 3 and 4 using both of KRAS4A and 4B. Binding of both KRAS4A and 4B to DX2 was increased by treatment with FTase inhibitor I (Supplementary Fig. 6b). The farnesylated KRAS4A and 4B showed weaker interaction with DX2 compared to the native counterparts (Fig. 3f and Supplementary Fig. 6c). DX2 increased KRAS4A and 4B levels in cytosol as well as in membrane fraction (Supplementary Fig. 6d). Besides, both of KRAS4A and 4B were similarly ubiquitinated by Smurf2 (Supplementary Fig. 8a) and binding with Smurf2 was blocked by DX2 in both forms (Supplementary Fig. 8h). Based on these results, DX2 appears to control both of KRAS4A and 4B with no apparent difference.

Q2-2) Based on their hypothesis and data listed in Figure 1-2, DX2 expression level and binding presumably only affect the protein level of KRAS in a GTP independent manner. It is somehow contradictory to their interactome data (figure 2A). In the setting of the experiment,

they identified DX2 interacting protein in EGF treated H460 cells. H460 cell line is known to be “KRAS mutant”, and EGF stimulation should only affect the RAS-GTP loading but not protein level. The authors then failed to address the effects of GTP binding in DX2-KRAS interaction in their protein-protein docking models. The lack of such data also weakens the importance of their finding, since KRAS-WT (which has higher binding affinity to GDP without stimulation) is known to have suppressive effect to the oncogenic activity of mutant KRAS. In addition, the protein level of KRAS (which they majorly focused on here) cannot completely translate to its oncogenic functions and activity.

Answer: To compare binding of DX2 to GTP-bound and -unbound conformation of KRAS, the KRAS4B structure in the final proposed model of DX2-KRAS complex was superimposed with the initial structure of GTP-bound conformation. The difference in C α -RMSD between proteins with and without GTP is about 1.7 Å, which is similar each other (Supplementary Fig. 5k), implying that DX2 could bind to KRAS in GTP-independent manner. Since DX2 would bind to cytosolic KRAS (GDP form), its binding to KRAS would not be affected by Q61H mutation of KRAS. The main reason for EGF-dependent increase of DX2 with KRAS appears to result from the changes of DX2 level and structure (Nat Chem Biol, 16:41, 2019). We share the reviewer’s comment on the translational meaning of KRAS level in tumorigenesis. In this work, we observed that increase of KRAS protein level enhances cell viability, proliferation and transformation in Supplementary Fig. 1f, 1g, 1h, 5r and 5s. In addition, increased levels of KRAS without oncogenic mutations have been reported in various cancers⁶⁻¹², implying an oncogenic potential of the increased KRAS level. Yet, our cancer cell and tissue analysis showed that the positive correlation of DX2 and KRAS levels is not 100% and thus may be applicable to a subset of cancer patients. This possibility was shown in the experimental results showing the cytotoxic activity of the DXI

compound to cancer cells expressing high levels of DX2 (Fig. 5g).

Q2-3) H460 is reported to be somehow KRAS independent (defined by Jeff Settleman's group; Cancer Cell 15, 489–500; June 2, 2009). It is therefore somehow surprising that their compound had such impact in H460 xenografts in Fig 5H. More cell lines with various KRAS-GTP level and/or dependency will strengthen the data.

Answer: We appreciate the reviewer's comments. To our knowledge, the Jeff Settleman's group reported that H460 cell line shows the KRAS-independency due to the lack of correlation of the KRAS levels with the increase of cleaved caspase3 and e-cadherin. We re-examined the above correlation using H460 (KRAS Q61H), A549 (KRAS G12S), HCC1588 (KRAS WT), and H1650 (KRAS WT), and found that the cell lines harboring KRAS WT show the correlation, but those harboring KRAS mutants as previously reported (Supplementary Fig. 12c, d). Interestingly, the decrease of KRAS level by the treatment of BC-DXI-32982 reduced phosphorylated ERK in all of the tested cell lines (Supplementary Fig. 12c, d), suggesting that inactivation of ERK would be a common reason for the decrease of cell proliferation resulting from the compound treatment. Following the reviewer's suggestion, we also tested the anti-tumor activity of BC-DXI-32982 in HCC1588 and A549 cells in addition to H460 cells (Fig. 5h, i) and observed the similar efficacy.

Q2-4) IHC in Fig 5H is hard to define the co-localization of KRAS and DX2. Also, it is known that there are no antibodies ideal for KRAS IHC. Yet, the information of the KRAS antibodies that they used here cannot be found in the manuscript. Positive and negative controls are therefore needed to validate their IHC.

Answer: We used anti-KRAS antibody (Novusbio Biologicals, NBP2-33579) for IHC. We

validated the capability of the antibody specifically recognizing KRAS using KRAS-knockout cell line *via* CRISPR/Cas9 (Abcam) and KRAS-overexpressed cell line (Supplementary Fig. 2d). These were described in Methods section (page 33, line 764). In the revised Supplementary Fig. 12h (showing immunohistochemistry of KRAS and DX2), we made the images of KRAS and DX2 levels in the different regions obtained from the same tissue. Thus, the DX2 and KRAS staining images would not overlap with each other.

Q2-5) A lot of important experiment details are missing. For example, the information of transgenic mice where DX2 expression can be induced by dox (line 60) is missing. No reference was provided either. On the similar note, the information of nanoluciferase-RAS constructs used in Figure 2b was not provided. Phosphorylation site of Akt in all western blots was not labeled.

Answer: We are thankful for the reviewer's comment. We added the full description of the indicated experiments to Methods section. Those include the construction of inducible DX2 knock-in mouse, preparation of its embryonic fibroblast cells and induction of DX2 by the treatment with doxycycline in Methods section (page 31, line 727). We also added the information of nanoluciferase-RAS isoform constructs used in Fig. 2b and c (page 35, line 817) and labelled the phosphorylation site of Akt, ERK and EGFR in all of the p-Akt, p-ERK and p-EGFR blots (Fig. 1a, 2e and 5e, and Supplementary Fig. 9a and 12c).

Referee #3:

Q3-1) The introduction is barely existent and does not provide adequate background information for the many aspects investigated in the result section. There are 2 sentences on KRAS, 2 sentences on DX2.

Answer: We added more background information of KRAS and DX2 in Introduction section with the significance of this research (page 3, line 41 ~ 90).

Q3-2) In the discussion, if statements refer to findings within the manuscript, the corresponding figure should be cited at the end of the sentence. If sentences refer to a previously described finding, the corresponding literature should be cited at the end of the sentence. Overall, the discussion could expand more on the context of the findings rather than repeating the result section and would benefit from language editing.

Answer: As reviewer's comments, we added the corresponding figure number at the end of the sentences and the expanded contents from our findings in Discussion section. We also edited the language of the entire paper by a professional editor.

Q3-3) A lot could be done to make it easier for readers to understand the figures. For example, Fig 4g – it is not clear that this is IP until one reads the figure legend and it is not clear that the time refers to time after EGF addition until one looks at Supplementary Figure 8a and b. What is meant by relative binding here? The same applies to several of the main figures where just looking at the figure, it is not clear what the main message, experimental system, or focus of the figure should be. Of course, figure legends are always necessary, but it would help a lot with comprehension if the figures would be more self-explanatory. Clearly, the authors put lots of thought and work into their experiments - making the data more accessible would help enormously for the community to appreciate their findings.

Answer: We are deeply thankful for the reviewer's advice. We made the original Fig. 4g as line graph to clearly show the EGF-dependent time course of the indicated protein pairs. In the revised figure, we displayed the results as a bar graph, showing the quantified levels of

the indicated protein pairs that were determined by immunoprecipitation assays. The maximum blot intensities obtained from immunoprecipitation of each protein pair were taken as 1, and the other blot intensities were divided by the maximum intensity values and presented as the relative values. This information was described in corresponding figure legend (page 50, line 1283). The meaning of all the figures were also improved for clear message. The abbreviations were also described in their first appearance.

Q3-4) Protein sizes are annotated inconsistently in western blots and gels, as are fold-changes. It should be noted how often experiments were repeated throughout and if technical or biological replicates were used. n numbers for animal experiments must be noted in the figure legends. There are no statistics on anything. It is appreciated how clean some of the results are, but still, statistics and quantification are necessary.

Answer: We annotated protein sizes of western blots in Source Data 2 including uncropped data. We also addressed biological replicates and n number for animal experiments in figure legends. For statistics, we displayed statistical significance in the plot of each figure and added the analysis method to Methods section (page 39, line 914).

Q3-5) The interactome of DX2 shown in Figure 2a contains many proteins. The method section does not mention if a fold-change over EV was used to determine true interactors over contaminants. The relative abundance in the corresponding table is < 1 for all interactors (DX2/EV) – if that is correct there is no enrichment over EV? Additionally, while KRAS is unarguably an important protein, it is far from the most enriched. Are the other interactions competing with KRAS for DX2 binding?

Answer: We are thankful for the reviewer's point. In fact, it was our mistake to label

“relative abundance”. Those values mean the fold change of DX2 over EV. Fold change value of the most frequently detected protein was set to “1”, and the remaining values were expressed in proportion. Thus, the exact meaning of the label is “Relative value of fold change” and we addressed above in Methods section (page 30, line 708) and changed the label in Figure and Results section, and Supplementary Table 2.

Since PRDX1 is most frequently detected as a potential interactor of DX2 in proliferative ontology, we tested whether PRDX1 would affect the binding between DX2 and KRAS under EGF signal by immunoprecipitation. Although PRDX1 could bind to DX2 independently of EGF, it did not hinder the EGF-induced interaction of DX2 and KRAS. We added these results to Supplementary Fig. 3d.

Q3-6) The predicted interaction interface of DX2/KRAS4B is mostly located in exon 3 – how would the insertion of exon 2 in AIMP2 disrupt the interaction to achieve the specificity that is shown for DX2 over AIMP2?

Answer: Since AIMP2 and DX2 would have the same structure in exon 3 and 4, the similar degree of KRAS binding was shown in both proteins as determined by *in vitro* pull-down assay (Supplementary Fig. 4d). However, AIMP2 is less accessible to KRAS (Fig. 2e) since AIMP2 is mainly sequestered in multi-tRNA synthetase complex (MSC) in contrast to DX2 that mainly exists as a free form³. To further validate this possibility, we performed gel filtration assay. As expected, AIMP2 and DX2 were mainly detected as MSC-bound and -unbound forms, respectively (Supplementary Fig. 4e). Thus, the preferred KRAS binding of DX2 over AIMP2 appears to result from the difference in cellular location, but not in structure. This point was described (page 8, line 180).

Q3-7) Is total ubiquitination affected by compound 32982? At least it should be shown that ubiquitination of other proteins are not affected by 32982 to confirm that 32982 is not globally upregulating ubiquitination.

Answer: We checked whether BC-DXI-32982 would affect total ubiquitination, and observed little effect on ubiquitination to global proteins (Supplementary Fig. 11f).

Q3-8) What is the proportion of endogenous DX2 to full-length AIMP2 in the cell lines that were used? Is there expected to be enough DX2 to stabilize KRAS?

Answer: To provide the answer to the reviewer's question, we checked the proportion of DX2, AIMP2, and KRAS in H460 cell lines by immunoblotting using a mixture of their specific antibodies and observed the results as shown below. Although immunoblotting does not quantitatively reflect the exact stoichiometry among them, DX2 appears to exist at the reasonable level to protect KRAS from degradation.

Q3-9) Are any KRAS mutations found in cancer that are predicted to localize to the interface of the KRAS/DX2 interaction that would stabilize/destabilize the interaction?

Answer: We found the disease-related mutations of KRAS at K5 and I36 residues that are critical for binding with DX2. K5E and I36M mutants were reported to be associated with gastric cancer and noonan syndrome^{13,14} although their pathological correlations and effects on the interaction of DX2 and KRAS are unclear. We mentioned this point in Discussion section (page 23, line 524).

Q3-10) Figure 2e, the DX2 blot looks odd as the background is a lot lighter on one side.

Answer: To further clarify the result shown in Fig. 2e, we repeated the same experiment and added the new results in the revised Fig. 2e

Q3-11) Fig 4g – why does the red line peak at 20 minutes when there is no time point measured and 10 and 30 minutes have the same relative binding? Same for 40 minute time point on the right panel, also red line.

Answer: The lines in Fig. 4g in original Figure were automatically drawn using the trend line in Prism (GraphPad). We changed the graph showing, not trend, the quantified values at the analyzed time points to more clearly deliver the meaning of graphs.

Q3-12) Fig 5h, IHC should contain a scale bar.

Answer: A scale bar was added to the revised Supplementary Fig. 12h.

Q3-13) Fig 11a – does 32982 display toxicity on non-KRAS dependent, untransformed cells, for example MEFs?

Answer: BC-DXI-32982 showed little toxic effect on cell viability in the untransformed MEF cells and cancer cells with low levels of DX2³ (Fig. 5g and Supplementary Fig. 12b).

Q3-14) How was the DX2 specific siRNA generated – does it span the exon 1 – exon 3 junction?

Answer: We used siRNA specifically targeting the junction of exon 1 and 3 of AIMP2 that was previously reported in “PLoS Genet 7, e1001351”, and added this information in

Methods section (page 25, line 581).

Q3-15) Supplementary Fig 2d: Why has the KRAS antibody been verified but not the DX2 antibody? The whole membrane should be shown here as unspecific bands with molecular weight other than the predicted molecular weight will give signal in IHC.

Answer: Validation of the antibody against DX2 for IHC was already reported in “Biomolecules 10: 820, 2020”. Nonetheless, we validated the DX2 antibody one more time in addition to validation of KRAS antibody in the revised Supplementary Fig. 2d. Uncropped images with molecular weight of blots for antibody validation were shown in Source Data 2.

Q3-16) Supplementary Figure 12e has a typo in FLAG. Figure 4c has a typo in Smurf2. Supplementary Fig 7d and e also have typos in Smurf2.

Answer: We thank the reviewer for careful checking. We corrected the typos in the revised figures.

Referee #4:

Q4-1) Both AIMP2 and its splicing variant DX2 contain the GST domain, which is important for the assembly of several components of the multi-synthetase complex (MSC) via AIMP2. Since this GST domain in DX2 appears to be important for the binding of the inhibitor 32982, it is very intriguing that it does not do the same with AIMP2. Although the authors briefly discussed this point in lines 359-360, it is not clear how this could be possible. Also, the cellular levels of components of the MSC are not affected by the inhibitor 32982, but this does not prove that the compound does not bind to AIMP2. The binding of 32982 to AIMP2 might occur without impacting the cell levels of proteins of the MSC. Another interesting

question is: Does the putative binding of 32982 prevents the assembly of MSC mediated by AIMP2?

Answer: As the reviewer pointed, both proteins contain the GST domain and thus are likely to interact with BC-DXI-32982. As expected, the compound bound to AIMP2 as well as DX2 in *in vitro* pull-down assay (Supplementary Fig. 13f). However, BC-DXI-32982 did not show the binding to endogenous AIMP2 (Supplementary Fig. 13g). To further confirm this possibility, we performed gel filtration assay. While AIMP2 was mainly detected in the MSC fraction, DX2 was present in the MSC-unbound fraction (Supplementary Fig. 4e). Thus, DX2 would be more accessible than AIMP2 to the compound. Besides, we also found that the compound did not affect the interaction of AIMP2 and KARS1 (lysyl-tRNA synthetase 1) that is known to pair within MSC⁴ (Supplementary Fig. 13h) and that of AIMP2 and p53 in nucleus⁵ (Supplementary fig. 13i), suggesting the specificity of the compound to the DX2-KRAS pair. We added this point (page 20, line 452).

Q4-2) Figure 1F lacks the units of the scale bar 0-10. What arbitrary units were chosen to assign the values 0 to 10?

Answer: The intensities of DX2 and KRAS bands in Western blots of the tested cells were measured. Among them, the quantitated maximum and minimum values were shown with the highest and lowest color intensity, respectively, and the remaining values were graded according to their relative values. The procedure was addressed in Methods section (page 34, line 787). In Figure 1f, the relative values graded from “0” to “10” were changed to those from “Min” to “Max”.

Q4-3) As a structural biologist, a general comment about the manuscript is that I would have

loved to see real structures of DX2 bound to KRAS, and how the binding of the inhibitor 32982 would block such interaction. Of course, the authors might agree with me that this is beyond this study, but it would certainly be very exciting to see this in be a future study. Indeed, the identification of the exact binding regions of KRAS to DX2, and the relative high affinities as measured by SPR, would suggest that crystallography studies are feasible. Nevertheless, I agree with the authors that the disordered nature of KRAS HVR would create hurdles for crystallization, thus I suggest to use short peptides of this region of KRAS to try co-crystallization with DX2.

Answer: We are thankful for the reviewer's kind comments. As the reviewer might imagine, we have already tried to get a co-crystal of the DX2-KRAS complex but failed to get it perhaps due to the disordered flexible regions of the two proteins. In the following works, we will keep trying to find a way to get the solid structural data.

Q4-4) Lines 147-150: the authors provide an interpretation about the significant line broadening a disappearance of peaks in the NMR spectra due to induced non-specific aggregation of DX2. Besides this explanation, it is also possible that there is binding and that this is in an intermediate exchange, likely in the μ s to ms timescale, which could not be detected in these NMR experiments.

Answer: We thank the reviewer for the reasonable comment. When a peptide binds to a protein with an intermediate exchange in μ s to ms timescale, the chemical shift perturbation of protein would result in significant signal broadening. If this is the case for the binding of HRAS HVR and RALA HVR, signal broadening would be observed at the binding site of DX2. Since the NMR signal broadening was observed throughout almost all residues of DX2, we assumed that non-specific aggregation of DX2 was induced by the peptides. However, as

reviewer pointed, we cannot exclude the possibility of exchange broadening due to non-specific intermediate exchange. We thus modified the text in Results section (page 10, line 207).

Q4-5) Lines 228-230: “Smurf2 is the ligase...” The authors did not investigate the other ligase (B-TRCP), therefore “the ligase” should be modified to “a ligase” ?. also in line 230, should not be written “...SMURF2 could function as the ligase ...”.

Answer: Following the reviewer’s comment, we changed the sentence from “suggesting that Smurf2 is the E3 ligase” to “suggesting that Smurf2 could function as an E3 ligase” (page 14, line 308) and from “Smurf2 could function the E3 ligase” to “Smurf2 could function as an E3 ligase” (page 14, line 310).

Q4-6) Please revise grammar of phrase in lines 103-106.

Answer: We changed the sentence from “To investigate the mechanisms by which DX2 enhances the stability of KRAS, DX2-interacting proteins in condition with EGF signal were identified by affinity purification using strep-tagged DX2, followed by liquid chromatography-mass spectrometry (LC-MS) (Supplementary Fig. 3a)” to “To investigate the mechanisms by which DX2 enhances the stability of KRAS, DX2-interacting proteins in the presence of epidermal growth factor (EGF) signal were enriched by affinity purification and identified by liquid chromatography-mass spectrometry (LC-MS) (Supplementary Fig. 3a)” (page 7, line 142).

Q4-7) Line 185: CRE1- mediated cleavage or RCE1? Please check this is correct.

Answer: We thank the reviewer for the careful checking. RCE1 is right and we corrected the

typo (page 12, line 261).

Q4-8) Phrase in lines 212-215 appears incomplete ?

Answer: To complete the exact meaning of the sentence, Word “Since” was removed in phrase (page 13, line 293).

Q4-9) Typo in panel 4c: there should be Smurf2 and not Smruf2.

Answer: We corrected it in Fig. 4c.

Q4-10) Typo in line 247: “...conditions were determined...”.

Answer: We corrected it (page 14, line 326). We are sorry for the frequent uncaredful typos.

Q4-11) Line 250: “...two different binding modes....”

Answer: To further clarify the meaning of the sentence, “Previous reports unveiled two different binding of DX2” was changed to “p14ARF and HSP70 have been previously reported as DX2-binding proteins” (page 15, line 328).

Q4-12) Please revise gramma of phrase at line 252-257.

Answer: We revised the text to “p14ARF and HSP70 have been previously reported as DX2-binding proteins. Thus, we checked how DX2 would encounter these proteins upon EGF signaling over time. In response to EGF signaling, DX2 first bound to HSP70 (in 5 min) and then to KRAS (in 10 min) in the cytosol. Binding of DX2 to p14ARF in the nucleus was also observed at 10 min after EGF signaling” (page 15, line 328).

Q4-13) Line 293: "...BC-DXI-32982 is a specific...."

Answer: We changed the sentence as suggested (page 16, line 370).

Q4-14) Figure 5g: bar scale 0-10 lacks units, μM ?

Answer: EC_{50} values were determined in each of the tested cell lines. In each type of cancers, the maximum and minimum EC_{50} values were represented as blue color with the highest and lowest intensity, respectively, and the rests were graded according to their relative values. This information was added to figure legend (page 51, line 1303).

Q4-15) Panel 5h (right) is not cited in the text ?

Answer: The immunohistochemistry data of DX2 and KRAS in the tumor tissues (shown in Fig. 5h, right panel) was moved to Supplementary Fig. 12h in the revised work and cited in the Results section (page 18, line 403).

Q4-16) please revise grammar of lines 329-330.

Answer: Since the additional data (Supplementary Fig. 13f-i) were added, the sentence in lines 329-330 was deleted.

Q4-17) Line 378: "...but these two residues..." ?

Answer: We changed the sentence from "but two residues" to "these residues" (page 22, line 517).

Q4-18) Line 381: remove "...and...".

Answer: To further clarify the meaning, we entirely changed the sentence (page 22, line 517

~ 519).

Q4-19) Line 383: correct “did not affected...”

Answer: We are sorry for the mistake. We corrected it to “... the binding of the two proteins was not affected by knockdown of RCE1 protein” (page 23, line 520).

Q4-20) Not sure it makes sense phrase at line 389-392.

Q4-21) Please revise phrase at lines 395-397.

Answer: To deliver our message more clearly, we completely revised phrase at line 389-397 (389-392 in Q4-20 and 395-397 in Q4-21) to “Uncontrolled proliferation and transformation of cancer cells requires the activations of oncogenes and inhibition of tumor suppressors. Upon EGF signaling, DX2 also moves to the nucleus, suppressing the tumor suppressor p14ARF. Considering the multifaceted roles of DX2 in tumorigenesis, targeting DX2 would provide an efficient way to control a broad spectrum of cancers.” (page 23, line 532 ~ 536). We also revised the phrase at 389-397 as in Q4-20 (page 23, line 532 ~ 536).

Referee #5:

Q5-1) Since human tissues are used, the compliance with the ethics and government regulations should be clearly stated in the manuscript.

Answer: We added the compliance with ethics and government regulations for using the human tissues in Methods section (page 33, line 770).

Q5-2) The in vivo animal studies were limited on only female mice. The SABV (sex as a

biological variable) should be considered.

Answer: To consider SABV (sex as a biological variable), we conducted *in vivo* study using both of male and female mice and found no difference in the chemical efficacy. We added the results to Supplementary Fig. 12m-q.

Q5-3) The *in vivo* animal studies of the inhibitor only used $n = 4$ per group, which is a little bit low. Also, more cell lines using KRAS wild type and mutant type should be conducted for the *in vivo* efficacy comparison for the inhibitor.

Answer: Following the reviewer's suggestion, we further evaluated the *in vivo* efficacy of BC-DXI-32982 by xenograft using KRAS wild type (HCC1588) and mutant (A549) cell lines, and observed similar anti-tumor efficacy of the compound regardless of mutation (Supplementary Fig. 12i-l). In this study, six mice were used per each group to obtain more reliable statistical results. We also repeated the mouse xenograft assay ($n = 6$) with H460 cells and added the results to Supplementary Fig. 12m-q.

Q5-4) The *in vivo* toxicity studies of the inhibitor should include the blood analysis and histology changes of various organs.

Answer: Following the reviewer's suggesting, we additionally performed *in vivo* toxicity studies on BC-DXI-32982, and added the results to the Results section (page 19, line 425) and Supplementary Note 1. Briefly, neither death nor macroscopic abnormalities were observed in the tested doses (10 and 50 mg/kg once a day), which are 10 times higher than those used for the *in vivo* efficacy test. In microscopic analysis, hepatocyte hypertrophy was observed in 10 and 50 mg/kg treated group. The values of monocyte and LUC were significantly increased and infiltration of mixed cells in lung was shown in the 50 mg/kg

treated group. However, the observed effects did not seem to be severe enough to consider them as a toxicity of the compound.

Q5-5) It is not clear how the dosages for testing the *in vivo* efficacy were chosen. The *in vivo* PK studies including the half-life and drug exposure, etc. should be analyzed.

Answer: We explained our strategy to select the doses for *in vivo* xenograft test with PK data (page 17, line 393), and added the PK results to Supplementary Note 1. We also analyzed the compound levels in the tumor tissues excised from mouse treated with BC-DXI-32982 and the compound levels were measured high enough for anti-tumor efficacy (Supplementary Fig. 12l, q, and Supplementary Note 1).

Q5-6) The binding affinity and IC50s should have +/- SD values for the variation.

Answer: We added “± SD” values to the results as pointed (Fig. 5b, and Supplementary Fig. 3f and 12a).

Q5-7) The off-target effects of BC-DXI-32982 are inadequately characterized. The target specificity of the reported inhibitor using a wide panel of drug targets (e.g., Eurofin drug target panel assays) should be pursued.

Answer: Following the reviewer’s comments, we performed Eurofin SafetyScreen44 Panel assay on BC-DXI-32982, and added the results to Supplementary Note 1. Among 44 panel assays, significant responses above 50% at 10 μM were noted in 5 primary assays including acetyl cholinesterase (66%), β2-adrenergic receptor (55%), L-type calcium channel (63%), cholecystokinin CCK1 (54%), and site 2 sodium channel (75%). Because these 5 targets are not related to ubiquitination-mediated degradation of KRAS, the anti-tumor efficacy of DXI

compound appears to result mainly from its inhibition of the DX2-KRAS interaction (page 18, line 418).

Q5-8) For the biotin tool compounds, the key moiety of quinoxaline in the parental inhibitor has been removed in the design of biotin-labeled tool molecules. Not sure how close these biotin tool compounds actually reflect the parental inhibitor in terms of the target engagement.

[REDACTED]

[REDACTED]

[REDACTED]

References

- 1 Kang, S. W. *et al.* Mammalian peroxiredoxin isoforms can reduce hydrogen peroxide generated in response to growth factors and tumor necrosis factor-alpha. *J Biol Chem* **273**, 6297-6302, doi:DOI 10.1074/jbc.273.11.6297 (1998).
- 2 Kadowaki, H. *et al.* Pre-emptive Quality Control Protects the ER from Protein Overload via the Proximity of ERAD Components and SRP. *Cell Rep* **13**, 944-956, doi:10.1016/j.celrep.2015.09.047 (2015).
- 3 Choi, J. W. *et al.* Cancer-Associated Splicing Variant of Tumor Suppressor AIMP2/p38: Pathological Implication in Tumorigenesis. *Plos Genet* **7**, doi:ARTN e100135110.1371/journal.pgen.1001351 (2011).
- 4 Kim, S., You, S. & Hwang, D. Aminoacyl-tRNA synthetases and tumorigenesis: more than housekeeping. *Nat Rev Cancer* **11**, 708-718, doi:10.1038/nrc3124 (2011).
- 5 Han, J. M. *et al.* AIMP2/p38, the scaffold for the multi-tRNA synthetase complex, responds to genotoxic stresses via p53. *P Natl Acad Sci USA* **105**, 11206-11211, doi:10.1073/pnas.0800297105 (2008).
- 6 Mita, H. *et al.* A novel method, digital genome scanning detects KRAS gene amplification in gastric cancers: involvement of overexpressed wild-type KRAS in downstream signaling and cancer cell growth. *Bmc Cancer* **9**, doi:Artn 19810.1186/1471-2407-9-198 (2009).
- 7 Hwang, K. T. *et al.* Prognostic Role of KRAS mRNA Expression in Breast Cancer. *J Breast Cancer* **22**, 548-561, doi:10.4048/jbc.2019.22.e55 (2019).
- 8 Rozhgar A. Khailany, M. S. a. M. O. Molecular Investigation of KRAS Gene in Breast Cancer Patients. *Journal of Biological Sciences* **19**, 323-327, doi:10.3923/jbs.2019.323.327 (2019).
- 9 Yarbrough, W. G. *et al.* Ras Mutations and Expression in Head and Neck Squamous-Cell Carcinomas. *Laryngoscope* **104**, 1337-1347 (1994).
- 10 Hoa, M., Davis, S. L., Ames, S. J. & Spanjaard, R. A. Amplification of wild-type K-ras promotes growth of head and neck squamous cell carcinoma. *Cancer Res* **62**, 7154-7156 (2002).
- 11 Mcdonald, J. S. *et al.* Immunohistochemical Detection of the H-Ras, K-Ras, and N-Ras Oncogenes in Squamous-Cell Carcinoma of the Head and Neck. *J Oral Pathol Med* **23**, 342-346, doi:DOI 10.1111/j.1600-0714.1994.tb00073.x (1994).
- 12 Rackley, B. *et al.* The level of oncogenic Ras determines the malignant transformation of Lkb1 mutant tissue in vivo. *Commun Biol* **4**, doi:ARTN 14210.1038/s42003-021-01663-8 (2021).
- 13 Lee, S. H. *et al.* BRAF and KRAS mutations in stomach cancer. *Oncogene* **22**, 6942-6945, doi:10.1038/sj.onc.1206749 (2003).
- 14 Duzkale, H. *et al.* A systematic approach to assessing the clinical significance of genetic variants. *Clin Genet* **84**, 453-463, doi:10.1111/cge.12257 (2013).

REVIEWER COMMENTS

Reviewer #1 (Remarks to the Author):

Summary: In this revision by Kim et al., the relationship between the protein AIMP2 (Aminoacyl tRNA synthase complex-interacting multifunctional protein 2, a tumor suppressor) and the RAS proteins is examined; specifically, DX2, which is a splicing variant of AIMP2 that lacks exon II and is upregulated in many cancers, is shown to bind with KRAS4b. The Authors describe (1) that DX2 enhances protein stability of KRAS; (2) that DX2 binds with specificity to KRAS4B (at a Kd of 151 nM by SPR), (3) the characterization of DX2-KRAS complex (4) that DX2 inhibits Smurf2-mediated ubiquitination of KRAS; (5) the identification and characterization of a DX2-KRAS inhibitor, BC-DX1-32982, which binds to DX2 with IC50 of 240 nM; and (6) Investigation of mode of action of BC-DX1-32982 (with a Kd of 480 nM to DX2 by a fluorescence based equilibrium binding assay). The authors use several experimental methods to interrogate the relationship between DX2 and KRAS including on purified protein, in cells, and in mouse models with xenografts, as well as performing a chemical screen to identify a lead molecule that disrupts the interactions between DX2 and KRAS. In addition, the authors use calculations to complement the experiments and make predictions that they tested experimentally. The authors appear to have addressed all my suggestions. The manuscript is greatly improved (and it was already a strong paper) by implementing the changes suggested by the other reviewers (and I hope my suggestions) and it now reads even better. I would recommend publication. The authors may consider the following suggestions/comments:

(1) I was now able to open the Supplementary Movie 2 (312385_0_video_352157_qsr7qr). It appears that the movie is time averaged or am I confused. I noticed the hydrogens on a methyl contracted into the carbon atoms I believe that this is an artifact of the time averaging there are some other artifacts that I also noticed. If it is so, the authors might consider stating that the movie is time averaged in the caption.

(2) I am assuming that the RMSD fits are self fits. The authors might consider specifying the fit in the caption for SI Figure 5. You could fit to the RAS structure and calculate the RMSD of the DX2 and vice versa to see the relative movement. Additionally, For the HVR movement, you could just fit to the G-domain of RAS to see the HVR movement. I am not sure if this additional RMSD analysis is needed for publication but would address my concern of about the stability of the complex model (see point 5).

(3) The authors mention catalytic inhibitor (page 3, line 63), catalytic pocket (page 4, line 65) about the G12C molecules. I am not sure this is completely accurate. From my understanding, the G12C molecules do not displace GTP/GDP. They bind in an allosteric pocket under the switch II region of the protein. I would suggest removing the word catalytic.

(4) On page 16 line 374 of the revised manuscript, the authors state that BI-2852 is a catalytic inhibitor. My understanding is this molecule binds in a pocket in between Switch I and Switch II – a pocket distal from the catalytic site. There is also some evidence the small molecule might induce dimer formation. It seems that most of the effect of this molecule is due to the interference of effector binding whether it is due to the formation of dimers or just through the binding of the small molecule is unclear. I would suggest removing the word catalytic.

(5) I am intrigued by the fact BI-2852 has little impact on binding of DX2. Particularly because it appears that some of the residues in interface 2 of the model of DX2-KRAS complex are shared with the binding pocket of BI-2852 (e.g. E37). I suspect that most of the binding is driven by the Hypervariable Region. This seems to be consistent with the discussion on page 8 lines 173-6. I wonder if it might be helpful to create an alternative model with just the HVR bound to DX2 without the G-domain interacting. The authors could consider generating additional models through docking. How stable would it be? This might be address through more simulations. I am not sure if additional simulations are needed for publication, but I am curious about the complex stability (see point 2 for discussion about the RMSD calculations).

(6) At first I thought that KRAS:MET188, DX2:LYS90 interaction as a salt-bridge in SI Table 3 was an error. But now I see that MET188 is the c-terminal residue, and its backbone carboxylate is engaging the lysine 90. Consider add a footnote to the table to specify that MET188 is the c-terminal to remove confusion.

Reviewer #2 (Remarks to the Author):

The authors have addressed the concerns.

Reviewer #3 (Remarks to the Author):

Overall, the manuscript is much improved and I appreciate the effort the authors put in. The following (minor) points should be addressed:

Starting the abstract with a disclaimer is at least unusual. I am sure the authors can find a more elegant way to introduce their findings.

The authors need to provide more background in the abstract – I appreciate that space is limited but protein names need to be given context (KRAS, Smurf2).

The introduction is much improved – the manuscript is much more accessible to a broader audience now.

Figure 1g. The description in line 132 “as found in cancer cell analysis” is very vague. The presentation of the result is unusual – I’m assuming the authors chose to not show a correlation as the staining was only classified as “high” and “low”. Just showing quadrants with numbers might be clearer than the pie chart.

Are 497 interactors realistic for DX2? It does seem like an unusually high number and should at least be discussed. Apart from fold-change, was any other measure employed to identify unspecific binders?

line 144: The relevance of EGF signaling is unclear – is a specific PTM state of KRAS expected upon EGF treatment? Are all KRAS cancers also EGF dependent (I don’t think so)? As this is a central point, clear context would be much appreciated.

line 284 has a typo: franesylation instead of farnesylation

line 300: this should be rephrased, “we decided to focus on Smurf2” for example, at the moment it sounds as if the authors “chose” Smurf2 as the ligase.

Line 517: I believe the phrasing “identified Smurf2 as one of the E3 ligases of KRAS” would be more correct given the findings on β TRCP?

Reviewer #4 (Remarks to the Author):

The authors have done a great job, have addressed all my points and have done even more very helpful experiments.

The manuscript looks absolutely fine in my opinion, all the errors seem to be appropriate.

Therefore I strongly support the publication of this manuscript.

Dr. Andrés Palencia
National Institute of Health and Medical Research (Inserm)
Institute for Advanced Biosciences (IAB), Grenoble, Inserm-UGA-CNRS
Group Leader of Structural Biology of Novel Drug Targets in Human Diseases
Webpage: <https://noveltargets-palencia.com>

Reviewer #5 (Remarks to the Author):

The authors were quite responsive to the previous comments and did an excellent job to address the concerns. The revised manuscript has been significantly improved. Most of my previous concerns have been addressed with significant amount of new data as suggested. It is acceptable along as other reviewers' concerns have been fully addressed.

Answer letter (Re: NCOMMS-21-19178A)

Reviewer #1:

Q1-1: I was now able to open the Supplementary Movie 2 (312385_0_video_352157_qsr7qr).

It appears that the movie is time averaged or am I confused. I noticed the hydrogens on a methyl contracted into the carbon atoms I believe that this is an artifact of the time averaging there are some other artifacts that I also noticed. If it is so, the authors might consider stating that the movie is time averaged in the caption.

Answer: Visualization of protein movement was difficult due to thermal fluctuation. Hence, the movie was recorded with trajectory smoothed option by averaging 5 frames of total ~2500 frames for 250 ns. We have added this statement to the legends of movie captions in Supplementary Information.

Q1-2: I am assuming that the RMSD fits are self fits. The authors might consider specifying the fit in the caption for SI Figure 5. You could fit to the RAS structure and calculate the RMSD of the DX2 and vice versa to see the relative movement. Additionally, For the HVR movement, you could just fit to the G-domain of RAS to see the HVR movement. I am not sure if this additional RMSD analysis is needed for publication but would address my concern of about the stability of the complex model (see point 5).

[REDACTED]

[REDACTED]

Q1-3: The authors mention catalytic inhibitor (page 3, line 63), catalytic pocket (page 4, line 65) about the G12C molecules. I am not sure this is completely accurate. From my understanding, the G12C molecules do not displace GTP/GDP. They bind in an allosteric pocket under the switch II region of the protein. I would suggest removing the word catalytic.

Answer: Following the reviewer's comment, we removed "catalytic" at the two places (page 3, line 65, and page 4, line 67).

Q1-4: On page 16 line 374 of the revised manuscript, the authors state that BI-2852 is a catalytic inhibitor. My understanding is this molecule binds in a pocket in between Switch I and Switch II – a pocket distal from the catalytic site. There is also some evidence the small molecule might induce dimer formation. It seems that most of the effect of this molecule is due to the interference of effector binding whether it is due to the formation of dimers or just through the binding of the small molecule is unclear. I would suggest removing the word catalytic.

Answer: We removed “catalytic” following the reviewer’s suggestion (page 16, line 370).

Q1-5: I am intrigued by the fact BI-2852 has little impact on binding of DX2. Particularly because it appears that some of the residues in interface 2 of the model of DX2-KRAS complex are shared with the binding pocket of BI-2852 (e.g. E37). I suspect that most of the binding is driven by the Hypervariable Region. This seems to be consistent with the discussion on page 8 lines 173-6. I wonder if it might be helpful to create an alternative model with just the HVR bound to DX2 without the G-domain interacting. The authors could consider generating additional models through docking. How stable would it be? This might be addressed through more simulations. I am not sure if additional simulations are needed for publication, but I am curious about the complex stability (see point 2 for discussion about the RMSD calculations).

[REDACTED]

[REDACTED]

Q1-6: At first I thought that KRAS:MET188, DX2:LYS90 interaction as a salt-bridge in SI Table 3 was an error. But now I see that MET188 is the c-terminal residue, and its backbone carboxylate is engaging the lysine 90. Consider add a footnote to the table to specify that MET188 is the c-terminal to remove confusion.

Answer: We added footnote to Supplementary table 3 to specify that MET188 is the c-terminal.

Reviewer #3:

Q3-1: Starting the abstract with a disclaimer is at least unusual. I am sure the authors can find a more elegant way to introduce their findings.

Answer: We changed the original sentence to more positive way as following. “Recent development of the chemical inhibitors specific to oncogenic KRAS (Kirsten Rat Sarcoma 2 Viral Oncogene Homolog) mutants revived much interest to control KRAS-driven cancers” (page 2, line 24).

Q3-2: The authors need to provide more background in the abstract – I appreciate that space is limited but protein names need to be given context (KRAS, Smurf2).

Answer: We added the full names to the proteins at their first appearance in abstract. Those include KRAS (Kirsten Rat Sarcoma 2 Viral Oncogene Homolog), AIMP2 (aminoacyl-tRNA synthetase-interacting multi-functional protein 2), and Smurf2 (SMAD Ubiquitination Regulatory Factor 2) (page 2, line 25, 27, and 31).

Q3-3: Figure 1g. The description in line 132 “as found in cancer cell analysis” is very vague.

The presentation of the result is unusual – I’m assuming the authors chose to not show a correlation as the staining was only classified as “high” and “low”. Just showing quadrants with numbers might be clearer than the pie chart.

Answer: Following the reviewers’ comment, we changed the pie chart to quadrants with numbers.

Q3-4: Are 497 interactors realistic for DX2? It does seem like an unusually high number and should at least be discussed. Apart from fold-change, was any other measure employed to identify unspecific binders?

Answer: In this work, we intentionally did not apply stringent pull-down condition not to lose any potential interactors of AIMP2-DX2. For this reason, we expect that many false positives may exist among the identified proteins. Instead, we applied comprehensive tests to validate the interaction of AIMP2-DX2 with KRAS is real.

Q3-5: line 144: The relevance of EGF signaling is unclear – is a specific PTM state of KRAS expected upon EGF treatment? Are all KRAS cancers also EGF dependent (I don’t think so)? As this is a central point, clear context would be much appreciated.

Answer: Post-translation modifications (PTMs) such as farnesylation and prenylation are known to be induced by growth factor receptors, especially epidermal growth factor (EGFR) to regulate membrane trafficking of KRAS. Other PTMs such as phosphorylation, acetylation, nitrosylation, and ubiquitination are also involved to control KRAS activity¹. However, the specific relationship between upstream signals and PTMs of KRAS is not yet fully determined.

While KRAS is activated by various growth factor receptor, epidermal growth factor (EGFR), fibroblast growth factor receptor (FGFR), vascular endothelial growth factor (VEGFR),

platelet derived growth factor receptor (PDGFR), and insulin-like growth factor 1 receptor (IGF-1R), the receptors responsible for the activation of KRAS depends on cell type and context^{2,3}. In this work, we just employed the EGF as a representative growth signal^{4,5}. We mentioned this point in Results section (page 7, line 145).

Q3-6: Line 284 has a typo: franesylation instead of farnesylation

Answer: We corrected the typo. Thanks for finding this (page 13, line 282).

Q3-7: Line 300: this should be rephrased, “we decided to focus on Smurf2” for example, at the moment it sounds as if the authors “chose” Smurf2 as the ligase.

Answer: The reviewer’s comment is reasonable. We thus changed the original sentence to “.. we decided to focus on Smurf2 ..” as suggested by the reviewer (page 13, line 299).

Q3-8: Line 517: I believe the phrasing “identified Smurf2 as one of the E3 ligases of KRAS” would be more correct given the findings on β TRCP?

Answer: We changed the sentence as the reviewer suggested (page 22, line 510).

References

- 1 Wang, W. H. *et al.* Post-translational modification of KRAS: potential targets for cancer therapy. *Acta Pharmacol Sin* **42**, 1201-1211, doi:10.1038/s41401-020-00542-y (2021).
- 2 Du, J., Yu, Y., Zhan, J. & Zhang, H. Targeted Therapies Against Growth Factor Signaling in Breast Cancer. *Adv Exp Med Biol* **1026**, 125-146, doi:10.1007/978-981-10-6020-5_6 (2017).
- 3 Snigdha Tiash, E. H. C. Growth factor receptors: promising drug targets in cancer. *Journal of Cancer Metastasis and Treatment* **1**, 190-200 (2015).
- 4 Kolch, W. Coordinating ERK/MAPK signalling through scaffolds and inhibitors. *Nat Rev Mol Cell Bio* **6**, 827-837, doi:DOI 10.1038/nrm1743 (2005).
- 5 Wee, P. & Wang, Z. Epidermal Growth Factor Receptor Cell Proliferation Signaling Pathways. *Cancers (Basel)* **9**, doi:10.3390/cancers9050052 (2017).

REVIEWERS' COMMENTS

Reviewer #1 (Remarks to the Author):

In this second revision by Kim et al., the relationship between the protein AIMP2 (Aminoacyl tRNA synthase complex-interacting multifunctional protein 2, a tumor suppressor) and the RAS proteins is examined; specifically, DX2, which is a splicing variant of AIMP2 that lacks exon II and is upregulated in many cancers, is shown to bind with KRAS4b. The Authors describe (1) that DX2 enhances protein stability of KRAS; (2) that DX2 binds with specificity to KRAS4B (at a K_d of 151 nM by SPR), (3) the characterization of DX2-KRAS complex (4) that DX2 inhibits Smurf2-mediated ubiquitination of KRAS; (5) the identification and characterization of a DX2-KRAS inhibitor, BC-DX1-32982, which binds to DX2 with IC_{50} of 240 nM; and (6) Investigation of mode of action of BC-DX1-32982 (with a K_d of 480 nM to DX2 by a fluorescence based equilibrium binding assay). The authors use several experimental methods to interrogate the relationship between DX2 and KRAS including on purified protein, in cells, and in mouse models with xenografts, as well as performing a chemical screen to identify a lead molecule that disrupts the interactions between DX2 and KRAS. In addition, the authors use calculations to complement the experiments and make predictions that they tested experimentally. The authors addressed my questions about RMSD fit, protein stability in the simulations and specifically the stability of the HVR domain with DX2 in their response letter addressing my last review—HVR (specially, the terminal CVIM motif) seems to bind stably to DX2 even without the G-domain. The authors have addressed all my comments, questions, and concerns, I recommend publication.